# An inference problem in a mismatched setting: a spin-glass model with Mattis interaction

**Francesco Camilli, Pierluigi Contucci and Emanuele Mingione**

Dipartimento di Matematica, Università di Bologna, Bologna, Italy

## Abstract

The Wigner spiked model in a mismatched setting is studied with the finite temperature Statistical Mechanics approach through its representation as a Sherrington-Kirkpatrick model with added Mattis interaction. The exact solution of the model with Ising spins is rigorously proved to be given by a variational principle on two order parameters, the Parisi overlap distribution and the Mattis magnetization. The latter is identified by an ordinary variational principle and turns out to concentrate in the thermodynamic limit. The solution leads to the computation of the Mean Square Error of the mismatched reconstruction. The Gaussian signal distribution case is investigated and the corresponding phase diagram is identified.



# 1 Introduction

The fruitful interplay between disordered Statistical Mechanics and high dimensional inference has a classical example in the well known equivalence between the Sherrington-Kirkpatrick model on the Nishimori line [1] and the Wigner spiked model (see [2] and references therein) with Rademacher prior. The first is the prototype of mean-field disordered systems with a special choice for the spin interaction distribution, whereas the second amounts to the problem of reconstructing a binary signal sent through a noisy Gaussian channel in the optimal setting when the receiver knows the distribution of the signal and the noise. The correspondence is based on the fact that the Shannon entropy of the observations that the receiver uses to retrieve the signal coincides, up to simple additive terms, with the free energy of the mentioned Statistical Mechanics model. The proof of such correspondence relies on Bayes rule and the gauge invariance property of such systems. From those one can also show that all these models fulfill a set of identities and correlation inequalities [3–5] that imply the peculiar feature of the emerging thermodynamics, known as *replica symmetry*, *i.e.* the system properties are fully encoded in a self-averaging quantity, the overlap [6–8].

The more general setting instead, referred to as *mismatched*, in which the receiver has only a guess of the signal distribution and/or does not know the strength of the noise is a new and rapidly growing research field [9–13].

In this paper we work with a fully mismatched Wigner spiked model, where the receiver has no *apriori* knowledge of the signal and tries to reconstruct it only through Ising spins. Instead of using a max-likelihood approach to estimate the signal, which would correspond to the search of the ground state of a given Hamiltonian, we choose a typical configuration of the system at finite temperature, or equivalently we adopt the receiver's posterior mean as the estimator. The emerging Statistical Mechanics model turns out to be the sum of an SK with a two-body mean-field Mattis interaction.

Our main result is the rigorous exact solution of such model described by the two natural order parameters represented by the overlap distribution and the Mattis magnetization. We show that, while the first obeys a functional variational principle of Parisi type, the second is obtained through a classical one dimensional optimization problem. The proof relies on the crucial property of self-averaging of the Mattis magnetization. When the signal distribution is Gaussian the phase space is investigated and a tricritical point is identified separating paramagnetic, glassy and ferromagnetic phases.

The paper is organized as follows. Section 2 contains the definitions and the main results from the Statistical Mechanics point of view. Section 3 briefly outlines the link between the inference problem and the mentioned model. Section 4 contains the mathematical proofs. Section 5 analyses in detail the phase diagram related to the case of Gaussian signal distribution. Finally, Section 6 collects conclusions and outlooks.

# 2 Definitions and Main Results

Consider a system of $N$ interacting Ising spins described by a Sherrington-Kirkpatrick Hamiltonian with external random *iid* magnetic fields and a further two body interaction of Mattis type induced by the same magnetic fields. More specifically, to each site $i = 1, \ldots, N$ we associate a spin $\sigma_i \in \{+1, -1\}$. The state of the system will be completely identified by the vector $\boldsymbol{\sigma} = (\sigma_1, \ldots, \sigma_N) \in \{+1, -1\}^N =: \Sigma_N$. Furthermore, we assume that the spins have a uniform

prior distribution, namely $\mathbb{P}(\sigma_i = +1) = 1/2$. The Hamiltonian of the model hereby studied is

$$H_N(\boldsymbol{\sigma}; \mu, \nu, \lambda) \equiv H_N(\boldsymbol{\sigma}) = -\sum_{i,j=1}^{N}\left(z_{ij}\sqrt{\frac{\mu}{2N}}\sigma_i\sigma_j + \frac{\nu}{2N}\sigma_i\sigma_j\xi_i\xi_j\right) - \lambda\sum_{i=1}^{N}\xi_i\sigma_i, \qquad (1)$$

with $\mu, \nu \geq 0$, $\lambda \in \mathbb{R}$, $z_{ij} \overset{\text{iid}}{\sim} \mathcal{N}(0,1)$ and $\xi_i \overset{\text{iid}}{\sim} P_\xi$ independent of the $z_{ij}$'s, where $P_\xi$ is any distribution such that $\mathbb{E}[\xi_1^4] < \infty$. The $z_{ij}$'s and $\xi_i$'s play the role of quenched disorder in this model. The model is going to be described by the couple of order parameters

$$q_N(\boldsymbol{\sigma}, \boldsymbol{\tau}) = \frac{1}{N}\boldsymbol{\sigma}\cdot\boldsymbol{\tau} = \frac{1}{N}\sum_{i=1}^{N}\sigma_i\tau_i, \quad m_N(\boldsymbol{\sigma}|\xi) = \frac{1}{N}\sum_{i=1}^{N}\sigma_i\xi_i, \qquad (2)$$

where $\boldsymbol{\sigma}, \boldsymbol{\tau} \in \Sigma_N$ and $\xi = (\xi_1, \ldots, \xi_N)$. In what follows we will refer to $m_N(\boldsymbol{\sigma}|\xi)$ as Mattis magnetization. One can now separate the three contributions in the Hamiltonian (1), thus obtaining

$$H_N(\boldsymbol{\sigma}) = -\sqrt{\mu}\sum_{i,j=1}^{N}\frac{z_{ij}}{\sqrt{2N}}\sigma_i\sigma_j - \frac{N\nu}{2}m_N^2(\boldsymbol{\sigma}|\xi) - N\lambda m_N(\boldsymbol{\sigma}|\xi), \qquad (3)$$

where an SK-like term

$$H_N^{SK}(\boldsymbol{\sigma}) := -\sum_{i,j=1}^{N}\frac{z_{ij}}{\sqrt{2N}}\sigma_i\sigma_j \qquad (4)$$

at temperature $\sqrt{\mu}$ is clearly recognizable. The Boltzmann-Gibbs average will be denoted by

$$\langle\cdot\rangle_N = \frac{1}{Z_N}\sum_{\boldsymbol{\sigma}\in\Sigma_N}(\cdot)\exp[-H_N(\boldsymbol{\sigma})], \quad Z_N = \sum_{\boldsymbol{\sigma}\in\Sigma_N}\exp[-H_N(\boldsymbol{\sigma})]. \qquad (5)$$

Due to the presence of the quenched disorder, Boltzmann-Gibbs averages are in general random quantities.

We define the random and quenched pressures of the model respectively as

$$p_N(\mu, \nu, \lambda) = \frac{1}{N}\log\sum_{\boldsymbol{\sigma}\in\Sigma_N}\exp\left[-\sqrt{\mu}H_N^{SK}(\boldsymbol{\sigma}) + \frac{N\nu}{2}m_N^2(\boldsymbol{\sigma}|\xi) + N\lambda m_N(\boldsymbol{\sigma}|\xi)\right], \qquad (6)$$

$$\bar{p}_N(\mu, \nu, \lambda) = \mathbb{E}p_N(\mu, \nu, \lambda), \qquad (7)$$

where the expectation in the latter is taken w.r.t. all the disorder: $\mathbb{E} \equiv \mathbb{E}_\xi\mathbb{E}_{\mathbf{Z}}$. For future convenience, we also introduce the quenched pressure of an SK model with random magnetic fields $\xi_i \overset{\text{iid}}{\sim} P_\xi$ and its limit:

$$\bar{p}_N^{SK}(\beta, h) := \frac{1}{N}\mathbb{E}\log\sum_{\boldsymbol{\sigma}\in\Sigma_N}\exp\left[-\beta H_N^{SK}(\boldsymbol{\sigma}) + h\sum_{i=1}^{N}\xi_i\sigma_i\right], \qquad (8)$$

$$\mathcal{P}(\beta, h) := \inf_{\chi\in\mathcal{M}_{[0,1]}}\mathcal{P}(\chi; \beta, h) = \lim_{N\to\infty}\bar{p}_N^{SK}(\beta, h), \qquad (9)$$

where $\mathcal{M}_{[0,1]}$ is the space of distributions over $[0,1]$ and $\mathcal{P}(\chi; \beta, h)$ is the Parisi functional [14–17] (see Sect. 4.1 for a synthetic description). The last limit exists by a super-additivity argument [18] and depends implicitly on the distribution $P_\xi$. The main result of this paper is the variational principle for the thermodynamic limit of (7).

**Theorem 1** (Variational solution). *If $\mathbb{E}[\xi_1^4] < +\infty$ then*

$$p_N(\mu, \nu, \lambda) \xrightarrow{L^2} \lim_{N \to \infty} \bar{p}_N(\mu, \nu, \lambda) =: p(\mu, \nu, \lambda) = \sup_{x \in \mathbb{R}} \varphi(x; \mu, \nu, \lambda), \tag{10}$$

*where*

$$\varphi(x; \mu, \nu, \lambda) := -\frac{\nu x^2}{2} + \mathcal{P}(\sqrt{\mu}, \nu x + \lambda). \tag{11}$$

From the form of the variational principle we can deduce also the differentiability properties of the limiting pressure that we have collected in the following

**Corollary 2.** *$p(\mu, \nu, \lambda)$ is $\lambda$-differentiable if and only if $\varphi(\cdot; \mu, \nu, \lambda)$ has a unique supremum point $x = \bar{x}(\mu, \nu, \lambda)$ and in that case*

$$\bar{x} = \frac{\partial}{\partial h} \mathcal{P}(\sqrt{\mu}, h) \Big|_{h = \nu \bar{x} + \lambda} = \lim_{N \to \infty} \mathbb{E}\langle m_N(\boldsymbol{\sigma}|\boldsymbol{\xi}) \rangle_N. \tag{12}$$

*$p(\mu, \nu, \lambda)$ is $\nu$-differentiable if and only if $\varphi(\cdot; \mu, \nu, \lambda)$ has at most two symmetric supremum points $\{\bar{x}, -\bar{x}\}$ and it holds*

$$\frac{\partial}{\partial \nu} p(\mu, \nu, \lambda) = \frac{\bar{x}^2}{2}. \tag{13}$$

*Let $\xi \sim P_\xi$ be centered and $\nu > 0$. If $\varphi(\cdot; \mu, \nu, \lambda = 0)$ has at most two symmetric supremum points $\{\bar{x}, -\bar{x}\}$ then $p(\mu, \nu, 0)$ is $\mu$-differentiable and it holds*

$$\frac{\partial}{\partial \mu} p(\mu, \nu, 0) = \frac{1}{4} \left( 1 - \int q^2 d\chi^*(\sqrt{\mu}, \nu \bar{x}; q) \right), \tag{14}$$

*where $\chi^*(\beta, h)$ denotes the unique Parisi measure solving the Parisi variational principle in* (9) *for $\beta = \sqrt{\mu}$, $h = \nu \bar{x}$.*

The proof of (10) relies on the adaptive interpolation introduced in inference in order to rigorously prove replica symmetric formulas [19–23] (see also [24, 25]). Within this technique the presence of a small perturbation in the Hamiltonian, appearing also in Proposition 3 below as $\epsilon$, plays a fundamental regularizing role. Intuitively, it allows us to avoid singularities that may occur in the thermodynamic limit in the vicinity of a possible phase transition. A similar model was studied in [26] where the author solves a Sherrington-Kirkpatrick model with an added ferromagnetic interaction, that can be derived from (1) setting $P_\xi = \delta_{\sqrt{J}}$ with $J > 0$ as the interaction strength. Notice moreover that the variational principle in (10) is one dimensional, as far as $x$ is concerned, suggesting thus the self-averaging of an order parameter to be identified with the Mattis magnetization as in (12). Indeed, the following concentration result holds.

**Proposition 3.** *Let $\epsilon \in [s_N, 2s_N]$ with $s_N = \frac{1}{2} N^{-\alpha}$, $\alpha \in (0, 1/2)$. Denote by $\langle \cdot \rangle_{N,y}$ the Boltzmann-Gibbs measure induced by the Hamiltonian $H_N(\boldsymbol{\sigma}; \mu, \nu, \lambda + y)$ for any $y \in \mathbb{R}$. Then*

$$\lim_{N \to \infty} \frac{1}{s_N} \int_{s_N}^{2s_N} d\epsilon \, \mathbb{E}\left\langle \left( m_N(\boldsymbol{\sigma}|\boldsymbol{\xi}) - \mathbb{E}\langle m_N(\boldsymbol{\sigma}|\boldsymbol{\xi}) \rangle_{N,\epsilon} \right)^2 \right\rangle_{N,\epsilon} = 0, \tag{15}$$

*for all $\mu, \nu \geq 0$ and $\lambda \in \mathbb{R}$.*

The proofs of Theorem 1, Corollary 2 and Proposition 3 can be found in Section 4.2 and require bounds on the fluctuations of $m_N(\boldsymbol{\sigma}|\boldsymbol{\xi})$.

## 2.1 The Gaussian case

Theorem 1 contains a variational representation for the thermodynamic limit of the quenched pressure density $p_N(\mu, \nu, \lambda)$ under mild assumption on the distribution of the family $\xi$. We should notice that despite the fact the variational problem is one dimensional, the potential $\varphi(x; \mu, \nu, \lambda)$ in (11) contains a very complicated object, namely the pressure of a SK model which is given by the Parisi formula. For this reasons it can be very hard in general to obtain analytical information on the solution of the above variational problem. For instance, an important question is when, once the potential $\varphi(x; \mu, \nu, \lambda)$ is evaluated at the optimal value for $x$, the Parisi term is solved by a non fluctuating order parameter, i.e. is replica symmetric. The purpose of this subsection is to obtain some detailed insights on the model by studying it on some analytically accessible case, in particular for a specific choice of the family $\xi$ that allows a quantitative description of the phase diagram. We choose the family $\xi$ to be $i.i.d$ centered Gaussian, $P_\xi = \mathcal{N}(0, a)$, and we set $\nu = \mu$ and $\lambda = 0$. The above choice for the parameters $\mu, \nu, \lambda$ and its link with high dimensional inference problems is discussed in Sect. 3. We will show that in this setting one can use the nice result in [27] on the sharpness of the de Almeida-Thouless line for Gaussian centered external magnetic fields for the SK model, to perform an in-depth analysis of the variational problem in Theorem 1. With a slight abuse of notation, we denote the corresponding quenched pressure by $\bar{p}_N(\mu, a)$. We show that it is possible to identify the regions in the phase plane $(\mu, a)$ where $\mathcal{P}$ defined in (9) can be replaced with its replica symmetric version, thus obtaining the following replica symmetric potential

$$\varphi_{RS}(x; \mu, a) := -\frac{\mu x^2}{2} + \frac{\mu(1 - q(x, \mu, a))^2}{4} + \mathbb{E} \log \cosh\left(z\sqrt{\mu q(x, \mu, a)} + \mu \xi x\right), \quad (16)$$

where $q(x, \mu, a)$ is uniquely defined, thanks to the Latala-Guerra lemma [28], by the consistency equation

$$q(x, \mu, a) = \mathbb{E} \tanh^2\left(z\sqrt{\mu q(x, \mu, a)} + \mu \xi x\right), \quad (17)$$

for any $x > 0$ and we extend it to $x = 0$ by continuity. The properties of $\varphi_{RS}$ are hereby collected:

**Proposition 4.** *The following properties hold:*

1. $\varphi_{RS}(-x; \mu, a) = \varphi_{RS}(x; \mu, a)$;

2. $\lim_{|x|\to\infty} \varphi_{RS}(x; \mu, a) = -\infty$;

3. *there exists a unique maximum point, up to reflection, $x = \bar{x}(\mu, a) \geq 0$ which is either 0 or satisfies*

$$q(\bar{x}(\mu, a), \mu, a) = 1 - \frac{1}{\mu a}; \quad (18)$$

4. *the solution of (18) exists and is unique if and only if*

$$a \geq \frac{1}{\mu(1 - q(0, \mu, 0))}. \quad (19)$$

   *The previous is always fulfilled if $a \geq 1/\mu$ and $a \geq 1$;*

5. *under the hypothesis (19) the solution to (18) is stable:*

$$\left.\frac{d^2 \varphi_{RS}(x; \mu, a)}{dx^2}\right|_{x=\pm\bar{x}(\mu,a)} < 0. \quad (20)$$

Finally, we give a sharp criterion to establish when the replica symmetric potential can be used to obtain the solution to the variational problem.

**Proposition 5.** *Define the function*

$$AT(\mu, a) := \mu \mathbb{E} \cosh^{-4}\left(z\sqrt{\mu q(\bar{x}(\mu, a), \mu, a)} + \mu \xi \bar{x}(\mu, a)\right). \tag{21}$$

*Then*

$$\lim_{N\to\infty} \bar{p}_N(\mu, a) = \sup_{x\in\mathbb{R}} \varphi_{RS}(x; \mu, a) \quad \textit{iff} \quad AT(\mu, a) \le 1. \tag{22}$$

The proofs of Propositions 4 and 5 can be found in Section 4.3. The previous results for Gaussian $\xi_i$'s and their consequences can be gathered together in the phase diagram in Figure 1 which will be studied in detail in the dedicated Section 5.

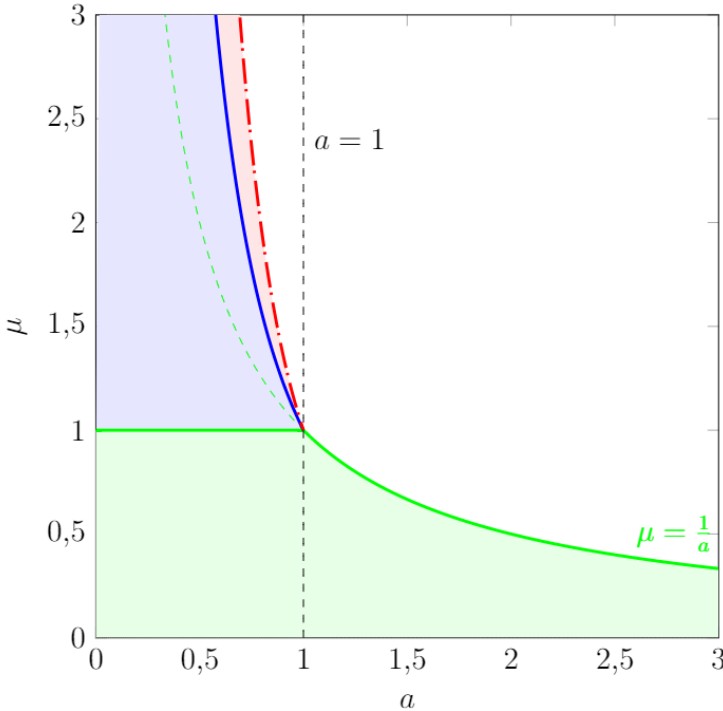

Figure 1: Model phase diagram when $\xi_i \overset{\text{iid}}{\sim} \mathcal{N}(0, a)$. Green region: fully paramagnetic phase, where the Parisi overlap distribution is a Dirac delta centered at 0 as well as the Mattis magnetization. White region: ferromagnetic, replica symmetric phase. Here, the Parisi overlap distribution is still a Dirac located according to (17) and (18). The distribution of the Mattis magnetization is instead a sum of two deltas centered at $\bar{x}$ and $-\bar{x}$ with $1/2$ coefficients, namely the solutions of (18). The model therefore turns out to be replica symmetric in the green and white areas. The blue region is delimited by $\mu = 1$ and the blue curve drawn (here only qualitatively) by (19), which is above the green dashed hyperbola $\mu = 1/a$. In this region the model is in a replica symmetry breaking phase, *i.e.* the Parisi distribution is no longer concentrated at a single point and $\bar{x}$ solves the more general variational principle (10). With reference to Proposition 5, the dash-dotted red line $AT(\mu, a) = 1$, here drawn qualitatively, must contain the entire RSB phase, hence it must lie above (or at most touch) the blue curve. The analogy with the SK model (see Remark 6 below) would suggest the presence of a mixed phase in the red region where $\bar{x} \ne 0$ and the overlap fluctuates.

# 3 Mismatched Setting in High Dimensional Statistical Inference

The Hamiltonian (1) with $\nu = \mu$ (which is not restrictive, one can reabsorb $\nu$ in the $\xi_i$'s) and $\lambda = 0$ can be derived also from a high dimensional inference problem, called Wigner spiked model in literature [19,29,30] (see also [31–34]), in a mismatched setting. In this problem the task is to recover a non negligible fraction of components of a high dimensional signal called *ground truth* $\xi$. The signal, in order to ease reconstruction, is sent in couples [1] through a channel which corrupts it with Gaussian noise $\mathbf{Z}$. The receiver will then get the message $\mathbf{y}$, *i.e.* the $N^2$ quantities

$$y_{ij} := \sqrt{\frac{\mu}{2N}}\xi_i\xi_j + z_{ij}, \quad z_{ij} \overset{\text{iid}}{\sim} \mathcal{N}(0,1). \tag{23}$$

He also knows how the observations are generated, namely he is aware of the law (23) and consequently of the conditional distribution

$$p_{\mathbf{Y}|\xi=\mathbf{x}}(\mathbf{y}) = \frac{\exp\left[-\frac{1}{2}\sum_{i,j=1}^N \left(y_{ij} - \sqrt{\frac{\mu}{2N}}x_i x_j\right)^2\right]}{(2\pi)^{N^2/2}}, \tag{24}$$

for some value $\mathbf{x}$. However, he does not know the distribution of the $\xi_i$'s and assumes them to be binary $\pm 1$ as the $\sigma_i$'s with equal prior probability. Thus, according to Bayes' rule, the posterior distribution used by the receiver is

$$P_{\xi|\mathbf{Y}=\mathbf{y}}(\boldsymbol{\sigma}) = \frac{\exp\left[-\frac{1}{2}\sum_{i,j=1}^N \left(y_{ij} - \sqrt{\frac{\mu}{2N}}\sigma_i\sigma_j\right)^2\right]}{2^N(2\pi)^{N^2/2}p_{\mathbf{Y}}(\mathbf{y})}, \tag{25}$$

where

$$p_{\mathbf{Y}}(\mathbf{y}) = \frac{1}{2^N}\sum_{\boldsymbol{\sigma}\in\Sigma_N}\frac{\exp\left[-\frac{1}{2}\sum_{i,j=1}^N \left(y_{ij} - \sqrt{\frac{\mu}{2N}}\sigma_i\sigma_j\right)^2\right]}{(2\pi)^{N^2/2}}. \tag{26}$$

A straightforward computation shows that the posterior distribution (25) can be rewritten as a random Boltzmann-Gibbs measure whose Hamiltonian is precisely $H_N(\boldsymbol{\sigma};\mu,\mu,0)$. To see this it is sufficient to compute the square at the exponent in (25) and reabsorb all the constants not depending on $\boldsymbol{\sigma}$ in the normalization.

It is important to stress that nor the posterior (25) neither the so called *evidence* (26) are correct, in the sense that there is a mismatch between the receiver's prior and $P_\xi$. The true distribution of the $y_{ij}$'s is instead

$$p_{\mathbf{Y}}^*(\mathbf{y}) = \int dP_\xi(\mathbf{x})\frac{\exp\left[-\frac{1}{2}\sum_{i,j=1}^N \left(y_{ij} - \sqrt{\frac{\mu}{2N}}x_i x_j\right)^2\right]}{(2\pi)^{N^2/2}}, \tag{27}$$

with $dP_\xi(\mathbf{x}) = \prod_{i=1}^N dP_\xi(x_i)$. With these notations one can proceed with the computation of the cross entropy density

$$\frac{1}{N}\mathcal{H}(p_{\mathbf{Y}}^*, p_{\mathbf{Y}}) = -\frac{1}{N}\int d\mathbf{y}\, p_{\mathbf{Y}}^*(\mathbf{y})\log p_{\mathbf{Y}}(\mathbf{y}), \tag{28}$$

a quantity that can be evaluated only by a third party observer aware of both $P_\xi$ and the mismatched prior. By inserting $p_\mathbf{Y}$ and the (23) in the (28) one obtains

$$
\begin{aligned}
\frac{1}{N}\mathcal{H}(p_\mathbf{Y}^*, p_\mathbf{Y}) &= -\frac{1}{N}\mathbb{E}_\mathbf{Z}\mathbb{E}_\xi \log \sum_{\sigma \in \Sigma_N} \frac{\exp\left[-\frac{1}{2}\sum_{i,j=1}^N \left(z_{ij} + \sqrt{\frac{\mu}{2N}}(\xi_i\xi_j - \sigma_i\sigma_j)\right)^2\right]}{2^N (2\pi)^{N^2/2}} \\
&= \frac{N}{2}\log 2\pi e + \frac{\mu}{4}\mathbb{E}_\xi^2[\xi_1^2] + \frac{\mu}{4} + O\left(\frac{1}{N}\right) + \log 2 - \frac{1}{N}\mathbb{E}_\mathbf{Z}\mathbb{E}_\xi \log \sum_{\sigma \in \Sigma_N} e^{-H_N(\sigma;\mu,\mu,0)}.
\end{aligned}
\tag{29}
$$

The first term is the Shannon entropy of the noise, whereas the last one is, up to a sign, the quenched pressure $\bar{p}_N(\mu, \mu, 0)$. In the optimal setting, namely when $P_\xi = (\delta_1 + \delta_{-1})/2$, (28) is just the entropy of the observations and the quantity $\frac{\mu}{4} + \frac{\mu}{4} + \log 2 - \bar{p}_N(\mu, \mu, 0)$ is the mutual information $\frac{1}{N}I(\mathbf{Y}, \xi)$ between the ground truth signal and the observations up to $O\left(\frac{1}{N}\right)$.

Using integration by parts it is straightforward to show that

$$
\frac{d}{d\mu}\frac{\mathcal{H}(p_\mathbf{Y}^*, p_\mathbf{Y})}{N} = \frac{\mathbb{E}_\xi^2[\xi_1^2] - 2\mathbb{E}\langle m_N^2(\sigma|\xi)\rangle_N + \mathbb{E}\langle q_N^2(\sigma, \tau)\rangle_N}{4} = \frac{1}{4N^2}\mathbb{E}\|\xi \otimes \xi - \langle \sigma \otimes \sigma\rangle_N\|_F^2,
\tag{30}
$$

where by $\langle f(\sigma, \tau)\rangle_N$ we mean the expectation w.r.t. the replicated Boltzmann-Gibbs measure $\langle \cdot \rangle_N^{\otimes 2}$, that averages over $\sigma$ and $\tau$ independently but with the same quenched disorder. The previous equation relates the cross entropy (28) to the theoretical expected Mean Square Error (MSE) in Frobenius' norm that the receiver would make using the Bayesian a posteriori estimator $\langle \sigma \otimes \sigma\rangle = (\langle \sigma_i\sigma_j\rangle)_{i,j=1}^N$ for the ground truth diad $\xi \otimes \xi = (\xi_i\xi_j)_{i,j=1}^N$. As intuition suggests, the estimation performed in the matched setting produces a MSE which is the smallest possible, therefore called Minimum Mean Square Error (MMSE). Namely there is no better estimator than the mean w.r.t the *true* posterior which in particular entails that the MSE in (30) is sub-optimal [9]. It is worth stressing again that the MSE (30) can be evaluated only by the aforementioned third party observer, since it derives directly from $\mathcal{H}(p_\mathbf{Y}^*, p_\mathbf{Y})$.

The MSE in the high dimensional limit can be evaluated using Theorem 1 and the following

**Lemma 6.** *Let $I$ be an open real interval, $\{g_n\}_{n\in\mathbb{N}}$ a sequence of differentiable functions defined on $I$ converging pointwise to a differentiabile function $g$. Suppose there exists a differentiable function $f$ on $I$ such that $\{g_n + f\}_{n\in\mathbb{N}}$ is a sequence of convex differentiable functions on $I$. Then $\lim_{N\to\infty} g_n'(x) = g'(x)$.*

*Proof.* The statement follows immediately from an application of Griffith's Lemma (see for instance [35], Lemma IV.6.3) to the sequence $\tilde{g}_n = g_n + f$. □

For our cross entropy density sequence, which is not convex due to the lack of Nishimori identities, one can prove that

$$
\frac{\tilde{\mathcal{H}}(p_\mathbf{Y}^*, p_\mathbf{Y})}{N} = \frac{\mathcal{H}(p_\mathbf{Y}^*, p_\mathbf{Y})}{N} - \mu \log \mu
\tag{31}
$$

is concave by a direct computation of its second derivative w.r.t. $\mu$. We leave the details of the computation to the interested reader. Hence the previous Lemma, under the hypothesis for $\mu(\nu)$-differentiability of $p(\mu, \nu, \lambda = 0)$ in Corollary 2, implies that

$$
\begin{aligned}
\lim_{N\to\infty}\frac{1}{4N^2}\mathbb{E}\|\xi \otimes \xi - \langle \sigma \otimes \sigma\rangle\|_F^2 &= \frac{d}{d\mu}\left[\frac{\mu}{4}\mathbb{E}_\xi^2[\xi_1^2] + \frac{\mu}{4} - p(\mu, \mu, 0)\right] \\
&= \frac{1}{4}\mathbb{E}_\xi^2[\xi_1^2] + \frac{1}{4} - \frac{\bar{x}^2}{2} - \frac{1}{2\sqrt{\mu}}\partial_\beta \mathcal{P}(\beta, \mu\bar{x})|_{\beta=\sqrt{\mu}} \\
&= \frac{1}{4}\mathbb{E}_\xi^2[\xi_1^2] + \frac{1}{4} - \frac{\bar{x}^2}{2} - \frac{1}{4}\left(1 - \int q^2 d\chi^*(\sqrt{\mu}, \mu\bar{x}; q)\right),
\end{aligned}
\tag{32}
$$

where we have replaced the derivative of the Parisi functional w.r.t. $\beta$ as prescribed by Corollary 2. We finally come up with

$$\lim_{N\to\infty}\frac{1}{4N^2}\mathbb{E}\|\xi\otimes\xi-\langle\boldsymbol{\sigma}\otimes\boldsymbol{\sigma}\rangle\|_F^2=\frac{1}{4}\mathbb{E}_\xi^2[\xi_1^2]-\frac{\bar{x}^2}{2}+\frac{1}{4}\int q^2 d\chi^*(\sqrt{\mu},\mu\bar{x};q)\,. \tag{33}$$

# 4  Proofs

The proofs of the results are presented by first introducing the necessary tools like the adaptive interpolation and the differentiability properties of the Parisi pressure.

## 4.1  Tools

In this section we show how to interpolate our model with a simple SK model with random *iid* magnetic fields [18] following an adaptive path [19]. The advantage of this approach is the possibility to confine the replica symmetry breaking phenomena in the SK part of the model which is exhaustively studied in the literature [15–17,28,36–42]. The ultimate purpose of the interpolation hereby illustrated is then to linearize the squared magnetization in the Hamiltonian.

The interpolating model is defined by means of the Hamiltonian

$$\begin{aligned}
H_N(t;\sigma):&=H_N(\sigma;\mu,(1-t)\nu,\lambda+R_\epsilon(t))\\
&=\sqrt{\mu}H_N^{SK}(\boldsymbol{\sigma})-(1-t)\frac{N\nu}{2}m_N^2(\boldsymbol{\sigma}|\xi)-(\lambda+R_\epsilon(t))Nm_N(\boldsymbol{\sigma}|\xi),
\end{aligned} \tag{34}$$

where

$$R_\epsilon(t)=\epsilon+\nu\int_0^t ds\, r_\epsilon(s),\quad \epsilon\in[s_N,2s_N],\quad s_N=\frac{N^{-\alpha}}{2}, \tag{35}$$

with $\alpha\in(0,1/2)$ and where the interpolating function $r_\epsilon$ will be suitably chosen (see Remark 3 below for instance). The related interpolating pressure is:

$$\bar{p}_N(t):=\bar{p}_N(\mu,(1-t)\nu,\lambda+R_\epsilon(t))=\frac{1}{N}\mathbb{E}_\xi\mathbb{E}_{\mathbf{z}}\log\sum_{\boldsymbol{\sigma}\in\Sigma_N}\exp[-H_N(t;\sigma)]\,. \tag{36}$$

As done in Proposition 3, the Boltzmann-Gibbs averages relative to (34) will be denoted by $\langle\cdot\rangle_{N,R_\epsilon(t)}$.

*Remark* 1. The interpolation strategy that we use in this work is profoundly different from the typical one of the statistical inference literature within the Bayes optimal setting [19, 20]. In that case one interpolates directly at the level of the channel, namely of (23), to compare it with a one body channel of the type $y_i=\sqrt{R_\epsilon(t)}\xi_i+z_i$ with $z_i\overset{\mathrm{iid}}{\sim}\mathcal{N}(0,1)$. Traveling along a trajectory that keeps an inferential interpretation ensures that the model is on the Nishimori line at any $t$ where all the precious properties of that line, identities and correlation inequalities, provide a crucial analytical tool to obtain a finite dimensional variational principle.

In the present case instead the structural complexity of the mismatched setting implies that we cannot count in the very first place on the Nishimori line properties nor on a global absence of fluctuations for the order parameters. The strategy to achieve the solution and overcome this difficulty is to build an interpolation scheme that, albeit not coming from a Gaussian channel of type (23), is able to isolate a pure SK part, described by the Parisi solution, plus a classical one dimensional variational principle.

*Remark* 2. In what follows, we will exploit the fact that the quenched pressure has bounded derivative in the external biases $\lambda$. Indeed, thanks to Cauchy-Schwartz inequality and to $\mathbb{E}[\xi_1^4] < \infty$ we have

$$\left|\frac{d}{d\lambda}\bar{p}_N(\mu, \nu, \lambda)\right| = |\mathbb{E}\langle m_N\rangle_N| \leq \frac{1}{N}\sum_{i=1}^{N}|\mathbb{E}\langle\sigma_i\xi_i\rangle_N| \leq \frac{1}{N}\sum_{i=1}^{N}\sqrt{\mathbb{E}[\xi_i^2]} = \sqrt{\mathbb{E}[\xi_1^2]} =: \sqrt{a}. \tag{37}$$

The previous bound holds for all $\mu, \nu \geq 0$ and $\lambda \in \mathbb{R}$. In particular, it holds for $\nu = 0$, namely for the SK model where this implies that $\bar{p}_N^{SK}(\cdot, h)$ is Lipschitz in $h$ and so will be its limit $\mathcal{P}(\cdot, h)$. Furthermore, since the interpolating model (34) is of the type (1) it inherits these Lipschitz properties on its quenched pressure (36).

**Proposition 7.** *The following sum rule holds:*

$$\bar{p}_N(\mu, \nu, \lambda) = \bar{p}_N^{SK}(\sqrt{\mu}, \lambda + R_\epsilon(1)) - \frac{\nu}{2}\int_0^1 dt[r_\epsilon^2(t) - \Delta_\epsilon(t)] + \mathcal{O}(s_N), \tag{38}$$

*where*

$$\Delta_\epsilon(t) := \mathbb{E}\left\langle\left(m_N(\boldsymbol{\sigma}|\xi) - r_\epsilon(t)\right)^2\right\rangle_{N,R_\epsilon(t)}. \tag{39}$$

*Proof.* Let us begin by computing the first derivative

$$\dot{\bar{p}}_N(t) = \mathbb{E}\left\langle-\frac{\nu}{2}m_N^2(\boldsymbol{\sigma}|\xi) + \nu r_\epsilon(t)m_N(\boldsymbol{\sigma}|\xi)\right\rangle_{N,R_\epsilon(t)} = \frac{\nu}{2}r_\epsilon^2(t) - \frac{\nu}{2}\Delta_\epsilon(t). \tag{40}$$

Remark 2 implies that

$$\bar{p}_N(0) = \bar{p}_N(\mu, \nu, \lambda) + \mathcal{O}(s_N); \tag{41}$$

$$\bar{p}_N(1) = \frac{1}{N}\mathbb{E}\log\sum_{\boldsymbol{\sigma}\in\Sigma_N}\exp\left[-\sqrt{\mu}H_N^{SK}(\boldsymbol{\sigma}) + (R_\epsilon(1) + \lambda)\sum_{i=1}^{N}\xi_i\sigma_i\right] \tag{42}$$
$$= \bar{p}_N^{SK}(\sqrt{\mu}, R_\epsilon(1) + \lambda).$$

An application of the fundamental theorem of calculus yields the result. $\square$

*Remark* 3. By looking at the remainder $\Delta_\epsilon(t)$ in the sum rule one may be led to choose the interpolating function as

$$r_\epsilon(t) = \mathbb{E}\langle m_N(\boldsymbol{\sigma}|\xi)\rangle_{N,R_\epsilon(t)}, \tag{43}$$

in order to apply Proposition 3 in some suitable form and make $\Delta_\epsilon(t)$ vanish in the thermodynamic limit. However, this can cause two issues. First, the extra bias $R_\epsilon(t)$ here introduced is not simply $\epsilon$ as required by Proposition 3 but a function of it, so one has to make sure that this does not interfere with the concentration. Second, in (43) $r_\epsilon(\cdot)$ appears implicitly on both sides of the equation. Nevertheless, the choice (43) can be formalized by means of the ODE

$$\dot{R}_\epsilon(t) = \nu\mathbb{E}\langle m_N(\boldsymbol{\sigma}|\xi)\rangle_{N,R_\epsilon(t)} =: G_N(t, R_\epsilon(t)), \quad R_\epsilon(0) = \epsilon, \tag{44}$$

which has always a solution by Cauchy-Lipschitz theorem because the velocity field $G_N$ is Lipschitz for fixed $N$ in the spatial coordinate $R_\epsilon$

$$\frac{\partial}{\partial R_\epsilon}G_N(t, R_\epsilon(t)) = \nu N\mathbb{E}\left\langle(m_N(\boldsymbol{\sigma}|\xi) - \langle m_N(\boldsymbol{\sigma}|\xi)\rangle_{N,R_\epsilon(t)})^2\right\rangle_{N,R_\epsilon(t)} \geq 0. \tag{45}$$

Furthermore, by Liouville's formula and the previous equation we know that the Jacobian $\partial_\epsilon R_\epsilon$ satisfies

$$\frac{\partial R_\epsilon(t)}{\partial \epsilon} = \exp\left\{ \int_0^t ds\, \frac{\partial G_N(s, R_\epsilon(s))}{\partial R_\epsilon} \right\} \geq 1. \tag{46}$$

Equations (44), (45) and (46) together provide a rigorous justification to the choice (43) and solve the aforementioned issues as it will be clear from the proof of Theorem 1 below.

Let us now turn to Corollary 2. For its proof we need to give more details on $\mathcal{P}(\beta, h)$. Consider the space of atomic probability measures on $[0, 1]$, denoted by $\mathcal{M}^d_{[0,1]}$, and define the Parisi functional

$$\chi \in \mathcal{M}^d_{[0,1]} \longmapsto \mathcal{P}(\chi; \beta, h) = \log 2 + \bar{\Phi}_\chi(0, h; \beta) - \frac{\beta^2}{2} \int_0^1 dq\, q\chi([0, q]). \tag{47}$$

$\bar{\Phi}_\chi$ is here introduced as an expectation: $\bar{\Phi}_\chi(s, h; \beta) = \mathbb{E}\Phi_\chi(s, h\xi; \beta), s \in [0, 1]$ where $\Phi_\chi$ solves the final value problem

$$\partial_s \Phi_\chi(s, y; \beta) = -\frac{\beta^2}{2} \left( \partial_y^2 \Phi_\chi(s, y; \beta) + \chi([0, s])(\partial_y \Phi_\chi(s, y; \beta))^2 \right), \tag{48}$$
$$\Phi_\chi(1, y; \beta) = \log \cosh y.$$

It is well known [15, 16] that for any $\chi_1, \chi_2 \in \mathcal{M}^d_{[0,1]}$

$$|\Phi_{\chi_1}(s, y; \beta) - \Phi_{\chi_2}(s, y; \beta)| \leq \frac{\beta^2}{2} \int_s^1 dq\, |\chi_1([0, q]) - \chi_2([0, q])|, \tag{49}$$

namely $\chi \mapsto \Phi_\chi$ is Lipschitz in the $L^1([s, 1], dq)$ norm. This allows us to extend the functional $\Phi_\chi$ to all the probability measures $\mathcal{M}_{[0,1]}$ with the prescription

$$\Phi_\chi := \lim_{n \to \infty} \Phi_{\chi_n}, \tag{50}$$

for any sequence $(\chi_n)_{n \geq 1}$ in $\mathcal{M}^d_{[0,1]}$ such that $\chi_n \longrightarrow \chi \in \mathcal{M}_{[0,1]}$ weakly.

We hereby collect the continuity and differentiability properties of $\bar{\Phi}_\chi$ and $\mathcal{P}(\chi; \cdot, \cdot)$.

**Proposition 8.** *Let $a := \mathbb{E}\xi^2$. The following hold:*

*i) $\bar{\Phi}_\chi$ (and $\mathcal{P}(\chi; \cdot, \cdot)$) can be continuously extended to $\mathcal{M}_{[0,1]}$ w.r.t. the weak convergence and*

$$\bar{\Phi}_\chi(s, h; \beta) := \lim_{n \to \infty} \bar{\Phi}_{\chi_n}(s, h; \beta) = \mathbb{E}\Phi_\chi(s, h\xi; \beta) \tag{51}$$

*for any sequence $(\chi_n)_{n \geq 1}$ in $\mathcal{M}^d_{[0,1]}$ such that $\chi_n \longrightarrow \chi \in \mathcal{M}_{[0,1]}$ weakly.*

*ii) $\chi \mapsto \bar{\Phi}_\chi$ is convex in $\mathcal{M}_{[0,1]}$.*

*iii) $\bar{\Phi}_\chi$ is twice h-differentible for any $\chi \in \mathcal{M}_{[0,1]}$ and*

$$|\partial_h \bar{\Phi}_\chi(s, h; \beta)| \leq \sqrt{a}, \quad 0 < \partial_h^2 \bar{\Phi}_\chi(s, h; \beta) \leq a. \tag{52}$$

*In particular it is convex in h.*

*iv) Consider a sequence $(\chi_n)_{n \geq 1}$ in $\mathcal{M}_{[0,1]}$ such that $\chi_n \longrightarrow \chi \in \mathcal{M}_{[0,1]}$ weakly. Then*

$$\partial_h \bar{\Phi}_{\chi_n} \longrightarrow \partial_h \bar{\Phi}_\chi. \tag{53}$$

*v)* *The function* $\mathcal{P}(\beta,h) = \inf_{\chi \in \mathcal{M}_{[0,1]}} \mathcal{P}(\chi;\beta,h)$ *is h-differentiable at any* $h \in \mathbb{R}$ *and*

$$\partial_h \mathcal{P}(\beta,h) = \partial_h \mathcal{P}(\chi^*(\beta,h);\beta,h) = \partial_h \bar{\Phi}_{\chi^*(\beta,h)}(0,h;\beta), \tag{54}$$

*where* $\chi^*(\beta,h)$ *is the unique distribution at which the infimum is attained and only the explicit dependence on h is taken into account.*

*Proof. i).* Consider $\chi_1, \chi_2 \in \mathcal{M}^d_{[0,1]}$. By (49)

$$|\bar{\Phi}_{\chi_1}(s,h;\beta) - \bar{\Phi}_{\chi_2}(s,h;\beta)| \le \frac{\beta^2}{2} \int_s^1 dq \, |\chi_1([0,q]) - \chi_2([0,q])|, \tag{55}$$

namely $\chi \mapsto \bar{\Phi}_\chi$ is Lipschitz too on $\mathcal{M}^d_{[0,1]}$. Therefore we perform a continuous extension to $\mathcal{M}_{[0,1]}$ obtaining a continuous functional with respect to the weak convergence. Furthermore, given a sequence $(\chi_n)_{n\ge 1}$ converging to $\chi \in \mathcal{M}_{[0,1]}$ weakly we have

$$|\bar{\Phi}_{\chi_n}(s,h;\beta) - \mathbb{E}\Phi_\chi(s,h\xi;\beta)| \le \frac{\beta^2}{2} \int_s^1 dq \, |\chi_n([0,q]) - \chi([0,q])| \longrightarrow 0, \tag{56}$$

by dominated convergence.

*ii).* The thesis immediately follows from *i)* and the main result in [42] that asserts the convexity of $\Phi_\chi$.

*iii).* By Proposition 2 in [42] the first two $y$-derivatives of $\Phi_\chi$ exist and are continuous, with $|\partial_y \Phi_\chi(s,y;\beta)| \le 1$, $C/\cosh^2 y \le \partial_y^2 \Phi_\chi(s,y;\beta) \le 1$ for some $C > 0$. Then, using Lagrange's mean value theorem and dominated convergence one can show that

$$\partial_h \bar{\Phi}_\chi(s,h;\beta) = \mathbb{E}\left[\xi \partial_y \Phi_\chi(s,h\xi;\beta)\right], \quad \partial_h^2 \bar{\Phi}_\chi(s,h;\beta) = \mathbb{E}\left[\xi^2 \partial_y^2 \Phi_\chi(s,h\xi;\beta)\right], \tag{57}$$

which implies (52) and in turn the convexity of $\bar{\Phi}_\chi$ in $h$.

*iv).* Since $\bar{\Phi}_\eta$ is convex in $h$ for any $\eta \in \mathcal{M}_{[0,1]}$, $\bar{\Phi}_{\chi_n}$ is a sequence of convex functions. Therefore, thanks to points and *i), ii)* and *iii)*

$$\lim_{n\to\infty} \partial_h \bar{\Phi}_{\chi_n} = \partial_h (\lim_{n\to\infty} \bar{\Phi}_{\chi_n}) = \partial_h \bar{\Phi}_\chi. \tag{58}$$

*v).* $\mathcal{P}(\beta,h)$ is convex in $h$ because it is the limit of a sequence of convex functions. Hence it is sufficient to prove that at any $h \in \mathbb{R}$ the sub-differential is single valued (as done for instance in [43]). For any fixed $\delta > 0$ and $b$ in the sub-differential the following holds

$$\frac{\mathcal{P}(\beta,h) - \mathcal{P}(\beta,h-\delta)}{\delta} \le b \le \frac{\mathcal{P}(\beta,h+\delta) - \mathcal{P}(\beta,h)}{\delta}. \tag{59}$$

Now, thanks to point *i)* and *ii)*, $\mathcal{P}(\chi;\beta,h)$ is also $\chi$-convex, thus it has a unique minimizer $\chi^*$, and it is continuous w.r.t. the weak convergence. Hence we can find a sequence of measures such that $\chi_n \longrightarrow \chi^*$ weakly and

$$\mathcal{P}(\chi_n;\beta,h) \le \mathcal{P}(\chi^*;\beta,h) + \frac{1}{n} = \mathcal{P}(\beta,h) + \frac{1}{n} \tag{60}$$

whilst it is obvious that $\mathcal{P}(\chi_n;\beta,h) \ge \mathcal{P}(\beta,h)$. Inserting these inequalities in (59) produces

$$-\frac{1}{n\delta} + \frac{\mathcal{P}(\chi_n;\beta,h) - \mathcal{P}(\chi_n;\beta,h-\delta)}{\delta} \le b \le \frac{1}{n\delta} + \frac{\mathcal{P}(\chi_n;\beta,h+\delta) - \mathcal{P}(\chi_n;\beta,h)}{\delta}. \tag{61}$$

Notice that $\partial_h^{1,2}\mathcal{P}(\chi_n;\beta,h)=\partial_h^{1,2}\bar{\Phi}_{\chi_n}(0,h;\beta)$ hence we can expand the Parisi functional up to the second order obtaining

$$-\frac{1}{n\delta}+\partial_h\mathcal{P}(\chi_n;\beta,h)-\frac{a\delta}{2}\le b\le\frac{1}{n\delta}+\partial_h\mathcal{P}(\chi_n;\beta,h)+\frac{a\delta}{2}\,,\qquad(62)$$

where we have used (52). Choose now $\delta=n^{-1/2}$ and then send $n\to\infty$. Finally, applying point *iv)* we conclude that the unique possible value for $b$ is $\partial_h\bar{\Phi}_{\chi^*}(0,h;\beta)$. $\qquad\square$

It was proved in [15, 16, 37] that the function $\mathcal{P}$ introduced in point *v)* of the above proposition is indeed the limit of (8).

## 4.2 Proofs of Theorem 1, Corollary 2 and Proposition 3

Both results need the $L^2$ convergence of the random pressure towards the limit of its expectations and a preliminary control on the fluctuations of the Mattis magnetization which are respectively contained in the two following lemmas.

**Lemma 9** (Self-averaging of the pressure). *If $\mathbb{E}[\xi_1^4]<\infty$ then*

$$\mathbb{E}\left[(p_N(\mu,\nu,\lambda)-\bar{p}_N(\mu,\nu,\lambda))^2\right]\le\frac{K(\mu,\nu,\lambda)}{N}\,,\quad K(\mu,\nu,\lambda)=C_1\mu+C_2\nu^2+C_3\lambda^2\,,\qquad(63)$$

*with $C_1,C_2,C_3>0$.*

*Proof.* The random pressure $p_N$ is a function of the random variables $(\mathbf{Z},\xi)$. For this proof we stress this dependency by writing $p_N(\mathbf{Z},\xi)$. Define $\mathbf{Z}^{(ij)}=(z_{12},z_{13},\ldots,z'_{ij},\ldots,z_{N,N-1})$ and $\xi^{(i)}=(\xi_1,\xi_2,\ldots,\xi'_i,\ldots,\xi_N)$ where $z'_{ij}\sim\mathcal{N}(0,1)$ and $\xi'_i\sim P_\xi$ are independent of anything else. Then, by Efron-Stein inequality

$$\mathbb{E}\left[(p_N(\mu,\nu,\lambda)-\bar{p}_N(\mu,\nu,\lambda))^2\right]\equiv\mathbb{V}[p_N(\mathbf{Z},\xi)]\le\frac{1}{2}\sum_{i,j=1}^N\mathbb{E}\left[\left(p_N(\mathbf{Z}^{(ij)},\xi)-p_N(\mathbf{Z},\xi)\right)^2\right]$$
$$+\frac{1}{2}\sum_{i=1}^N\mathbb{E}\left[\left(p_N(\mathbf{Z},\xi^{(i)})-p_N(\mathbf{Z},\xi)\right)^2\right].\qquad(64)$$

Let us focus on the terms in the first sum. By Lagrange's mean value theorem we have that there exists a $\tilde{z}_{ij}\in(\min(z_{ij},z'_{ij}),\max(z_{ij},z'_{ij}))$ such that

$$\left(p_N(\mathbf{Z}^{(ij)},\xi)-p_N(\mathbf{Z},\xi)\right)^2=\left(\frac{\partial p_N}{\partial z_{ij}}\bigg|_{\tilde{z}_{ij}}\right)^2(z_{ij}-z'_{ij})^2$$
$$=\left(\frac{1}{N}\sqrt{\frac{\mu}{2N}}\langle\sigma_i\sigma_j\rangle_{N,\tilde{z}_{ij}}\right)^2(z_{ij}-z'_{ij})^2\le\frac{\mu}{2N^3}(z_{ij}-z'_{ij})^2\,,\qquad(65)$$

where by $\langle\cdot\rangle_{N,\tilde{z}_{ij}}$ we mean the Boltzmann-Gibbs measure where $z_{ij}$ has been replaced with $\tilde{z}_{ij}$ in the Hamiltonian (1). In a really similar fashion we estimate the second set of terms. Again,

let $\tilde{\xi}_i \in (\min(\xi_i, \xi'_i), \max(\xi_i, \xi'_i))$

$$
\left( p_N(\mathbf{Z}, \xi^{(i)}) - p_N(\mathbf{Z}, \xi) \right)^2 = \left( \frac{\partial p_N}{\partial \xi_i} \bigg|_{\tilde{\xi}_i} \right)^2 (\xi_i - \xi'_i)^2
$$

$$
= \left[ \frac{v}{N^2} \left( \sum_{j \neq i, 1}^{N} \xi_j \langle \sigma_i \sigma_j \rangle_{N, \tilde{\xi}_i} + \tilde{\xi}_i \right) + \frac{\lambda}{N} \langle \sigma_i \rangle_{N, \tilde{\xi}_i} \right]^2 (\xi_i - \xi'_i)^2 \quad (66)
$$

$$
= \left[ \frac{v}{N^2} \left( \sum_{j \neq i, 1}^{N} \xi_j \langle \sigma_i \sigma_j \rangle_{N, \tilde{\xi}_i} + \tilde{\xi}_i \right) + \frac{\lambda}{N^2} \sum_{j=1}^{N} \langle \sigma_i \rangle_{N, \tilde{\xi}_i} \right]^2 (\xi_i - \xi'_i)^2 .
$$

Notice that in the square bracket we have an overall sum of $2N$ terms. We can use Jensen's inequality to bring the square inside the sums. The last line of the previous is bounded by

$$
2N \left[ \frac{v^2}{N^4} \left( \sum_{j \neq i, 1}^{N} \xi_j^2 \langle \sigma_i \sigma_j \rangle_{N, \tilde{\xi}_i}^2 + \tilde{\xi}_i^2 \right) + \frac{\lambda^2}{N^4} \sum_{i=1}^{N} \langle \sigma_i \rangle_{N, \tilde{\xi}_i}^2 \right] (\xi_i - \xi'_i)^2 , \quad (67)
$$

whence, exploiting the fact that $\tilde{\xi}_i^2 \leq \max(\xi_i^2, \xi_i'^2) \leq \xi_i^2 + \xi_i'^2$

$$
\left( p_N(\mathbf{Z}, \xi^{(i)}) - p_N(\mathbf{Z}, \xi) \right)^2 \leq \frac{2}{N^3} \left[ v^2 \left( \sum_{j=1}^{N} \xi_j^2 + \xi_i'^2 \right) + N \lambda^2 \right] (\xi_i - \xi'_i)^2 . \quad (68)
$$

From the previous equation one can clearly see that $\xi$ appears at most at the 4th power on the r.h.s. Hence, thanks to the hypothesis, inserting the estimates (65) and (68) into (64) we get the claimed inequality. $\qquad \square$

**Lemma 10.** *Let $y \in [y_1, y_2]$, $\delta \in (0, 1)$ and denote by $\langle \cdot \rangle_{N, y}$ the Boltzmann-Gibbs expectation associated to the Hamiltonian $H_N(\sigma; \mu, v, \lambda + y)$. Then*

$$
\mathbb{E} \left\langle \left( m_N(\sigma | \xi) - \langle m_N(\sigma | \xi) \rangle_{N, y} \right)^2 \right\rangle_{N, y} = \frac{1}{N} \frac{d^2}{dy^2} \bar{p}_N(\mu, v, \lambda + y), \quad (69)
$$

$$
\mathbb{E} \left[ \left( \langle m_N(\sigma | \xi) \rangle_{N, y} - \mathbb{E} \langle m_N(\sigma | \xi) \rangle_{N, y} \right)^2 \right] \leq \frac{12 K(\mu, v, |\lambda| + |y| + 1)}{\delta^2 N}
$$
$$
+ 8\sqrt{a} \frac{d}{dy} \left[ \bar{p}_N(\mu, v, \lambda + y + \delta) - \bar{p}_N(\mu, v, \lambda + y - \delta) \right], \quad (70)
$$

*with $a := \mathbb{E} \xi_1^2$.*

*Proof.* The concentration property (70) can be obtained from the self-averaging and the convexity properties of the pressure density, proved in Lemma 9, using a well-know argument in spin glass theory [44, 45]. The version of that argument applied here is analogous to the one appearing in [46]. In order to lighten the notation we neglect subscripts in the brackets for this proof. (69) follows from a simple computation of the second derivative on the r.h.s. Let us skip directly to (70). It is easy to see that both $p_N$ and $\bar{p}_N$ are convex in the external biases $\lambda$. We first evaluate the difference

$$
\left| \frac{d}{dy} \left[ p_N(\mu, v, \lambda + y) - \bar{p}_N(\mu, v, \lambda + y) \right] \right| = \left| \langle m_N \rangle - \mathbb{E} \langle m_N \rangle \right| . \quad (71)
$$

The difference between two convex differentiable functions can be bounded (see Lemma 3.2 in [17]) from above as follows

$$
\left| \frac{d}{dy} \left[ p_N(\mu, \nu, \lambda + y) - \bar{p}_N(\mu, \nu, \lambda + y) \right] \right| \leq \frac{1}{\delta} \sum_{u = y \pm \delta, \, y} |p_N(\mu, \nu, \lambda + u) - \bar{p}_N(\mu, \nu, \lambda + u)|
$$
$$
+ \frac{d}{dy} \left( \bar{p}_N(\mu, \lambda + y + \delta) - \bar{p}_N(\mu, \lambda + y - \delta) \right), \quad (72)
$$

for any $\delta > 0$. For our purposes, it is sufficient to restrict ourselves to $\delta \in (0,1)$. By squaring both sides, averaging w.r.t. the disorder and using Jensen's inequality we get

$$
\mathbb{E}\left[ (\langle m_N \rangle - \mathbb{E}\langle m_N \rangle)^2 \right] \leq \frac{4}{\delta^2} \sum_{u = y \pm \delta, \, y} \mathbb{E}\left[ (p_N(\mu, \nu, \lambda + u) - \bar{p}_N(\mu, \nu, \lambda + u)^2 \right]
$$
$$
+ 4 \left[ \frac{d}{dy} \left( \bar{p}_N(\mu, \nu, \lambda + y + \delta) - \bar{p}_N(\mu, \nu, \lambda + y - \delta) \right) \right]^2. \quad (73)
$$

By Lemma 9, each of the three terms in the first sum of the previous equation can be bounded by $K(\mu, \nu, |\lambda| + |y| + 1)/N$ and this explains the first term in (70). Concerning the second, notice that the derivative in the square brackets is positive thanks to the convexity of $\bar{p}_N$ and bounded as seen in Remark 2. The previous considerations imply that

$$
\left[ \frac{d}{dy} \left( \bar{p}_N(\mu, \nu, \lambda + y + \delta) - \bar{p}_N(\mu, \nu, \lambda + y - \delta) \right) \right]^2 \leq
$$
$$
2\sqrt{a} \left[ \frac{d}{dy} \left( \bar{p}_N(\mu, \nu, \lambda + y + \delta) - \bar{p}_N(\mu, \nu, \lambda + y - \delta) \right) \right], \quad (74)
$$

which concludes the proof. □

We start with Proposition 3 that is a direct consequence the previous Lemma.

*Proof of Proposition 3.* For future convenience we introduce the notation $\mathbb{E}_\epsilon[\cdot] = \frac{1}{s_N} \int_{s_N}^{2s_N} (\cdot)$. We first decompose the quenched variance

$$
\mathbb{E}\left\langle \left( m_N(\boldsymbol{\sigma}|\xi) - \mathbb{E}\langle m_N(\boldsymbol{\sigma}|\xi) \rangle_{N,\epsilon} \right)^2 \right\rangle_{N,\epsilon} = \mathbb{E}\left\langle \left( m_N(\boldsymbol{\sigma}|\xi) - \langle m_N(\boldsymbol{\sigma}|\xi) \rangle_{N,\epsilon} \right)^2 \right\rangle_{N,\epsilon}
$$
$$
+ \mathbb{E}\left[ \left( \langle m_N(\boldsymbol{\sigma}|\xi) \rangle_{N,\epsilon} - \mathbb{E}\langle m_N(\boldsymbol{\sigma}|\xi) \rangle_{N,\epsilon} \right)^2 \right]. \quad (75)
$$

The first term in the r.h.s. of the previous equation is the contribution due to the thermal fluctuations in the model, whilst the second one is due to the disorder.

*Thermal fluctuations:* Consider (69) with $y \equiv \epsilon \in [s_N, 2s_N]$ and take the expectation $\mathbb{E}_\epsilon$ of both sides:

$$
\Delta_T := \mathbb{E}_\epsilon \mathbb{E}\left\langle \left( m_N(\boldsymbol{\sigma}|\xi) - \langle m_N(\boldsymbol{\sigma}|\xi) \rangle_{N,\epsilon} \right)^2 \right\rangle_{N,\epsilon} = \frac{1}{N s_N} \int_{s_N}^{2s_N} d\epsilon \frac{d^2}{d\epsilon^2} \bar{p}_N(\mu, \nu, \lambda + \epsilon). \quad (76)
$$

Now, recalling that the derivatives of the pressure are bounded (see (37)) we immediately conclude that

$$
\Delta_T = \mathcal{O}\left( \frac{1}{N s_N} \right). \quad (77)
$$

*Disorder fluctuations:* Analogously take (70) with $y \equiv \epsilon \in [s_N, 2s_N]$ and average w.r.t. $\epsilon$ on both sides. Considering that $\epsilon \leq 1$ we have

$$
\begin{aligned}
\Delta_D &:= \mathbb{E}_\epsilon \mathbb{E}\left[ \left( \langle m_N(\boldsymbol{\sigma}|\boldsymbol{\xi}) \rangle_{N,\epsilon} - \mathbb{E}\langle m_N(\boldsymbol{\sigma}|\boldsymbol{\xi}) \rangle_{N,\epsilon} \right)^2 \right] \\
&\leq \frac{12 K(\mu, \nu, |\lambda| + 2)}{\delta^2 N} + \frac{8\sqrt{a}}{s_N} \int_{s_N}^{2s_N} d\epsilon \frac{d}{d\epsilon} \left[ \bar{p}_N(\mu, \nu, \lambda + \epsilon + \delta) - \bar{p}_N(\mu, \nu, \lambda + \epsilon - \delta) \right].
\end{aligned}
\tag{78}
$$

The last integral can be explicitly computed and then bounded by $4\delta\sqrt{a}$ thanks to Lagrange's mean value theorem and (37). Hence

$$
\Delta_D = \mathcal{O}\left( \frac{1}{\delta^2 N} + \frac{\delta}{s_N} \right),
\tag{79}
$$

which is optimized when $\delta = (s_N/N)^{1/3}$ (consistently with $\delta \in (0,1)$). This choice leads to

$$
\Delta_D = \mathcal{O}\left( \frac{1}{s_N^{2/3} N^{1/3}} \right) = \mathcal{O}\left( N^{\frac{2\alpha-1}{3}} \right).
\tag{80}
$$

The latter and (77) both vanish in the $N \to \infty$ limit for $\alpha \in (0, 1/2)$. $\qquad\square$

We are finally ready for the proof of Theorem 1.

*Proof of Theorem 1.* The variational principle is proven by means of two bounds that match in the thermodynamic limit. The lower bound follows from the classical sum rule combined with the positivity of the square. The upper bound is obtained with the adaptive interpolation method (see [20] for a nice introduction to this method). For the sake of clarity we consider each of them separately and then we prove (12).

*Lower bound:* Let us consider the sum rule (38) with the choice $r_\epsilon(t) = x$ constant in $t$. Furthermore observe that the remainder $\Delta_\epsilon(t)$ is always positive, so we discard it at the expense of an inequality:

$$
\bar{p}_N(\mu, \nu, \lambda) \geq \bar{p}_N^{SK}(\sqrt{\mu}, \lambda + \epsilon + \nu x) - \frac{\nu x^2}{2} + \mathcal{O}(s_N).
\tag{81}
$$

As explained in Remark 2 $\bar{p}_N^{SK}$ is Lipschitz in its second entry. This allows us to reabsorb the perturbation $\epsilon$ into $\mathcal{O}(s_N)$. By sending $N \to \infty$ one obtains the bound

$$
\liminf_{N\to\infty} \bar{p}_N(\mu, \nu, \lambda) \geq -\frac{\nu x^2}{2} + \mathcal{P}(\sqrt{\mu}, \nu x + \lambda),
\tag{82}
$$

which is uniform in $x$. We can optimize it by taking the $\sup_{x\in\mathbb{R}}$ on the r.h.s.

*Upper bound:* From (69) we see that any quenched pressure of the type (7) is convex in its third entry. Then, starting from the sum rule (38) we can use Jensen's inequality on the SK quenched pressure to obtain an upper bound

$$
\bar{p}_N(\mu, \nu, \lambda) \leq \mathcal{O}(s_N) + \int_0^1 dt \left[ -\frac{\nu r_\epsilon^2(t)}{2} + \bar{p}_N^{SK}(\sqrt{\mu}, \lambda + \epsilon + \nu r_\epsilon(t)) \right] + \frac{\nu}{2} \int_0^1 dt \, \Delta_\epsilon(t).
\tag{83}
$$

As done in the lower bound, we throw the dependence on $\epsilon$ in $\bar{p}_N^{SK}$ into $\mathcal{O}(s_N)$ and use Guerra's uniform bound $\bar{p}_N^{SK} \leq \mathcal{P}$ [15]:

$$
\begin{aligned}
\bar{p}_N(\mu, \nu, \lambda) &\leq \mathcal{O}(s_N) + \int_0^1 dt \left[ -\frac{\nu r_\epsilon^2(t)}{2} + \mathcal{P}(\sqrt{\mu}, \lambda + \nu r_\epsilon(t)) \right] + \frac{\nu}{2} \int_0^1 dt \, \Delta_\epsilon(t) \\
&\leq \mathcal{O}(s_N) + \sup_{x\in\mathbb{R}} \varphi(x; \mu, \nu, \lambda) + \frac{\nu}{2} \int_0^1 dt \, \Delta_\epsilon(t).
\end{aligned}
\tag{84}
$$

The only remaining dependency on the interpolation path is in $\Delta_\epsilon(t)$. To make the two bounds match we have to make sure the remainder vanishes in the limit. Hence, as suggested in Remark 3, we choose $r_\epsilon(\cdot)$ as in (43). At this point we can decompose $\Delta_\epsilon(t)$ as done in the proof of Proposition 3

$$
\Delta_\epsilon(t) = \mathbb{E}\Big\langle \big(m_N(\boldsymbol{\sigma}|\boldsymbol{\xi}) - \langle m_N(\boldsymbol{\sigma}|\boldsymbol{\xi})\rangle_{N,R_\epsilon(t)}\big)^2 \Big\rangle_{N,R_\epsilon(t)}
$$
$$
+ \mathbb{E}\Big[\big(\langle m_N(\boldsymbol{\sigma}|\boldsymbol{\xi})\rangle_{N,R_\epsilon(t)} - \mathbb{E}\langle m_N(\boldsymbol{\sigma}|\boldsymbol{\xi})\rangle_{N,R_\epsilon(t)}\big)^2\Big]. \tag{85}
$$

Let us first bound the $\epsilon$-average of the first term on the r.h.s. Using (69) and the inequality (46) on the Jacobian we get

$$
\frac{1}{s_N}\int_{s_N}^{2s_N} d\epsilon\, \mathbb{E}\Big\langle \big(m_N(\boldsymbol{\sigma}|\boldsymbol{\xi}) - \langle m_N(\boldsymbol{\sigma}|\boldsymbol{\xi})\rangle_{N,R_\epsilon(t)}\big)^2 \Big\rangle_{N,R_\epsilon(t)}
$$
$$
= \frac{1}{Ns_N}\int_{s_N}^{2s_N} d\epsilon\, \frac{d^2}{dy^2}\bar{p}_N(\mu,(1-t)v,\lambda+y)\Big|_{y=R_\epsilon(t)} \tag{86}
$$
$$
\leq \frac{1}{Ns_N}\int_{R_{s_N}(t)}^{R_{2s_N}(t)} dy\, \frac{d^2}{dy^2}\bar{p}_N(\mu,(1-t)v,\lambda+y) = \mathcal{O}\Big(\frac{1}{Ns_N}\Big),
$$

where the last equality follows from the bound on derivatives (37).

For the second term in the r.h.s. of (85) we use (70) and take its $\epsilon$-average:

$$
\frac{1}{s_N}\int_{s_N}^{2s_N} d\epsilon\, \mathbb{E}\Big[\big(\langle m_N(\boldsymbol{\sigma}|\boldsymbol{\xi})\rangle_{N,R_\epsilon(t)} - \mathbb{E}\langle m_N(\boldsymbol{\sigma}|\boldsymbol{\xi})\rangle_{N,R_\epsilon(t)}\big)^2\Big] \tag{87}
$$
$$
\leq \mathcal{O}\Big(\frac{1}{N\delta^2}\Big) + \frac{8\sqrt{a}}{s_N}\int_{s_N}^{2s_N} d\epsilon\, \frac{d}{dy}[\bar{p}_N(\mu,(1-t)v,\lambda+y+\delta) - \bar{p}_N(\mu,(1-t)v,\lambda+y-\delta)]\Big|_{y=R_\epsilon(t)}.
$$

Now, thanks again to inequality (46) and to the fact that the derivative of the square bracket is positive the integral in the previous equation can be bounded by

$$
\int_{R_{s_N}(t)}^{R_{2s_N}(t)} dy\, \frac{d}{dy}[\bar{p}_N(\mu,(1-t)v,\lambda+y+\delta) - \bar{p}_N(\mu,(1-t)v,\lambda+y-\delta)] \leq 4\sqrt{a}\delta. \tag{88}
$$

The last inequality follows from an application of the mean value theorem and (37). Equations (86), (87) and (88) together imply that

$$
\mathbb{E}_\epsilon[\Delta_\epsilon(t)] = \mathcal{O}\Big(\frac{1}{Ns_N} + \frac{1}{N\delta^2} + \frac{\delta}{s_N}\Big), \tag{89}
$$

that vanishes in the thermodynamic limit for $\delta = (s_N/N)^{1/3}$ and $s_N = 1/2 N^{-\alpha}$ with $\alpha \in (0,1/2)$ as seen for Proposition 3. With this information, we take the $\epsilon$-average on both sides of (84) and by Fubini's Theorem and dominated convergence we have

$$
\limsup_{N\to\infty} \bar{p}_N(\mu,v,\lambda) \leq \sup_{x\in\mathbb{R}} \varphi(x;\mu,v,\lambda). \tag{90}
$$

The two bounds, together with Lemma 9, conclude the proof of the variational principle (10). $\qquad\square$

*Remark* 4. The upper bound in the proof of (10) can also be obtained by adapting the elegant technique used in [26]. We opted instead for a proof that explicitly identifies the physical meaning of the vanishing distance between the upper and lower bounds in terms of the fluctuation of the order parameter. Such crucial thermodynamic property (Proposition 3) holds independently of the solution and it is at the origin of the (ordinary) variational principle in (10).

*Remark* 5. Since for $v > 0$ $\mathcal{P}(\beta, h)$ is $h$-Lipschitz we have

$$\lim_{|x| \to \infty} \varphi(x; \mu, v, \lambda) = -\infty, \tag{91}$$

therefore the supremum of $\varphi(\cdot; \mu, v, \lambda)$ will be attained at a finite $\bar{x} \in \mathbb{R}$. Furthermore the necessary condition for $\bar{x}$ to be a maximum point is

$$\bar{x} = \partial_h \mathcal{P}(\sqrt{\mu}, h)|_{h = v\bar{x} + \lambda}, \tag{92}$$

that in turn implies $\bar{x} \in [-\sqrt{a}, \sqrt{a}]$ by (52). Hence one can take the supremum only over $[-\sqrt{a}, \sqrt{a}]$.

*Proof of Corollary 2.*

$\lambda$-*differentiability:* Set $\Omega(\mu, v, \lambda) := \mathrm{argmax}_{[-\sqrt{a}, \sqrt{a}]} \varphi(\cdot; \mu, v, \lambda)$. Then, since $p(\mu, v, \lambda)$ is convex in $\lambda$, by Danskin's theorem (see [26] for instance) we have that the left and right derivatives satisfy respectively

$$\frac{d}{d\lambda_-} p(\mu, v, \lambda) = \min_{x \in \Omega(\mu, v, \lambda)} \frac{\partial}{\partial \lambda} \varphi(x; \mu, v, \lambda) = \min_{x \in \Omega(\mu, v, \lambda)} \frac{\partial}{\partial h} \mathcal{P}(\sqrt{\mu}, h)|_{h = vx + \lambda}, \tag{93}$$

$$\frac{d}{d\lambda_+} p(\mu, v, \lambda) = \max_{x \in \Omega(\mu, v, \lambda)} \frac{\partial}{\partial \lambda} \varphi(x; \mu, v, \lambda) = \max_{x \in \Omega(\mu, v, \lambda)} \frac{\partial}{\partial h} \mathcal{P}(\sqrt{\mu}, h)|_{h = vx + \lambda}. \tag{94}$$

If $\Omega(\mu, v, \lambda)$ is a singleton then $p(\mu, v, \lambda)$ is differentiable. Conversely, suppose that $p(\mu, v, \lambda)$ is differentiable and that there are at least two distinct values $x_1, x_2 \in \Omega(\mu, v, \lambda)$, $x_1 < x_2$. Then we have

$$\frac{d}{d\lambda_-} p(\mu, v, \lambda) \leq \frac{\partial}{\partial h} \mathcal{P}(\sqrt{\mu}, h)|_{h = vx_1 + \lambda} = x_1 < x_2$$

$$= \frac{\partial}{\partial h} \mathcal{P}(\sqrt{\mu}, h)|_{h = vx_2 + \lambda} \leq \frac{d}{d\lambda_+} p(\mu, v, \lambda), \tag{95}$$

that is a contradiction.

Assume now that there is a unique maximum point $\bar{x}$. Thanks to the convexity of the sequence $\bar{p}_N$ in $\lambda$ and Danskin's theorem we can write

$$\lim_{N \to \infty} \mathbb{E} \langle m_N(\boldsymbol{\sigma}|\boldsymbol{\xi}) \rangle_N = \lim_{N \to \infty} \frac{d}{d\lambda} \bar{p}_N(\mu, v, \lambda) = \frac{d}{d\lambda} p(\mu, v, \lambda) = \frac{\partial}{\partial \lambda} \varphi(\bar{x}; \mu, v, \lambda)$$

$$= \frac{\partial}{\partial \lambda} \mathcal{P}(\sqrt{\mu}, v\bar{x} + \lambda) = \frac{\partial}{\partial h} \mathcal{P}(\sqrt{\mu}, h)|_{h = v\bar{x} + \lambda} = \bar{x}, \tag{96}$$

where it is understood that only explicit dependence on $\lambda$ is taken into account when the partial derivative is taken.

$v$-*differentiability:* The proof relies on Danskin's theorem and is a straightforward consequence of that of Proposition 2 in [26].

$\mu$-*differentiability at $\lambda = 0$:* Notice that when $\xi$ is centered then $\varphi(x; \mu, v, 0)$ is symmetric in $x$. The result then follows easily again from Danskin's Theorem (as in Theorem 2 in [26]) and from the differentiability properties w.r.t. $\beta = \sqrt{\mu}$ of the Parisi pressure in Theorem 14.11.6 of [16] and [43].

$\square$

### 4.3 Proof of Proposition 4 and Proposition 5

*Proof of Proposition 4.* The fact that $\varphi_{RS}$ is an even function of $x$ follows directly from the symmetry of the random variable $\xi$. By Remark 5, when $|x| \to \infty$, the term $-\mu x^2/2$ is dominant in (16) bringing $\varphi_{RS}$ to $-\infty$. As a consequence the maximum point(s) of $\varphi_{RS}$ are critical point(s). The vanishing derivative condition yields

$$\frac{d\varphi_{RS}}{dx} = -\mu x + \mu \mathbb{E}\xi \tanh\left(z\sqrt{\mu q(x,\mu,a)} + \mu\xi x\right) = -\mu x + \mu^2 ax(1-q(x,\mu,a)) = 0, \quad (97)$$

that is

$$x = 0 \quad \text{or} \quad q(x,\mu,a) = 1 - \frac{1}{\mu a}. \quad (98)$$

Since the function is $q(x,\mu,a)$ increasing for $x \geq 0$, the positive solution $\bar{x}(\mu,a)$ of (18) exists and is unique up to reflection if and only if

$$\lim_{x \to 0^+} q(x,\mu,a) \leq 1 - \frac{1}{\mu a}, \quad (99)$$

which is equivalent to (19).

Consider now $a \geq 1$. If $1/a \leq \mu \leq 1$ we have $q(0,\mu,0) = 0$, thus (19) is clearly satified. Furthermore, it turns out that

$$\mu > 1 \quad \Rightarrow \quad AT(\mu,0) > 1. \quad (100)$$

In fact for $a = 0$ (1) reduces to an SK model with zero external magnetic field at temperature $\sqrt{\mu}$. Fix $\mu > 1$ and assume that $AT(\mu,0) \leq 1$. Then for any $\epsilon > 0$ by the monotonicity of $q(x,\mu,a)$

$$\mu \mathbb{E}\cosh^{-4}\left(z\sqrt{\mu q(\epsilon,\mu,1) + \epsilon^2\mu^2}\right) < AT(\mu,0) \leq 1. \quad (101)$$

[27] implies that the Parisi measure is $\chi^*(\sqrt{\mu},\epsilon\mu) = \delta_{q(\epsilon,\mu,1)}$ and $\chi^*(\sqrt{\mu},\epsilon\mu) \longrightarrow \delta_{q(0,\mu,0)}$ weakly. Since $\mathcal{P}(\beta,h)$ is continuous in $h$ and the Parisi functional $\mathcal{P}(\chi;\beta,h)$ is weakly continuous we have that

$$\mathcal{P}(\sqrt{\mu},0) = \lim_{\epsilon \to 0} \mathcal{P}(\sqrt{\mu},\epsilon\mu) = \mathcal{P}(\delta_{q(0,\mu,0)};\sqrt{\mu},0). \quad (102)$$

However for $\mu > 1$ we have $\mathcal{P}(\sqrt{\mu},0) < \mathcal{P}(\delta_{q(0,\mu,0)};\sqrt{\mu},0)$ thus the latter is a contradiction coming from the assumption $AT(\mu,0) \leq 1$. This proves (100). Hence

$$1 < \mu\mathbb{E}\cosh^{-4}(z\sqrt{\mu q(0,\mu,0)}) \leq \mu\mathbb{E}\cosh^{-2}(z\sqrt{\mu q(0,\mu,0)}) = \mu(1-q(0,\mu,0)), \quad (103)$$

from which, when $a \geq 1$,

$$q(0,\mu,0) < 1 - \frac{1}{\mu} \leq 1 - \frac{1}{\mu a}. \quad (104)$$

Finally, the solution to (18) is stable w.r.t. the optimization, indeed

$$\frac{d^2\varphi_{RS}(x;\mu,a)}{dx^2}\bigg|_{x=\bar{x}(\mu,a)} = -\mu + \mu^2 a(1-q(x,\mu,a)) - \mu^2 a\bar{x}(\mu,a)\frac{dq}{dx}(\bar{x}(\mu,a),\mu)$$

$$= -\mu^2 a\bar{x}(\mu,a)\frac{dq}{dx}(\bar{x}(\mu,a),\mu) < 0 \quad (105)$$

thanks to the monotonicity of $q(x,\mu,a)$. The result for $x = -\bar{x}(\mu,a)$ follows by symmetry. $\quad\square$

*Proof of Proposition 5.* By proposition 4 there exists a unique (non negative) maximum point $\bar{x}(\mu, a)$ of $\varphi_{RS}(x; \mu, a)$. Given $(\mu, a) \in \mathbb{R}_{\geq 0} \times \mathbb{R}_{\geq 0}$ we introduce the following subset of the real line:

$$RS(\mu, a) = \left\{ x \in \mathbb{R} \,|\, \mu \, \mathbb{E} \cosh^{-4} \left( z \sqrt{\mu \, q(x, \mu, a) + \mu^2 x^2 a} \right) \leq 1 \right\}. \tag{106}$$

Clearly by definition $AT(\mu, a) \leq 1 \iff \bar{x}(\mu, a) \in RS(\mu, a)$. We start assuming that $\bar{x}(\mu, a) \in RS(\mu, a)$. Let us denote by $\varphi(x; \mu, a)$ the variational potential (11) specialized in the current setting, namely with $\nu = \mu$, $\lambda = 0$ and $\xi \sim \mathcal{N}(0, a)$. From (82) we already know that

$$\liminf_{N \to \infty} \bar{p}_N(\mu, a) \geq \varphi(x; \mu, a) \tag{107}$$

uniformly on $x$. Hence we can optimize (107) only over the region $RS(\mu, a)$ obtaining the lower bound:

$$\liminf_{N \to \infty} \bar{p}_N(\mu, a) \geq \sup_{x \in RS(\mu, a)} \varphi(x; \mu, a). \tag{108}$$

The choice of restricting the supremum to the region $RS(\mu, a)$ allows us to replace in (108) the function $\varphi$ with its its replica symmetric version $\varphi_{RS}$. Indeed again by [27] the AT condition is sufficient for the validity of the the replica symmetric solution of the SK model. Then from (108) one gets the lower bound

$$\liminf_{N \to \infty} \bar{p}_N(\mu, a) \geq \sup_{x \in RS(\mu, a)} \varphi_{RS}(x; \mu, a). \tag{109}$$

For the upper bound we can exploit the fact that the pressure of the SK model is always bounded from above by the replica symmetric one [15]. Hence from the upper bound (90) we get

$$\limsup_{N \to \infty} \bar{p}_N(\mu, a) \leq \sup_{x} \varphi_{RS}(x; \mu, a) = \sup_{x \in RS(\mu, a)} \varphi_{RS}(x; \mu, a), \tag{110}$$

where the last equality follows from the assumption $\bar{x}(\mu, a) \in RS(\mu, a)$. Summarising we just proved that

$$AT(\mu, a) \leq 1 \implies \lim_{N \to \infty} \bar{p}_N(\mu, a) = \sup_{x \in RS(\mu, a)} \varphi_{RS}(x; \mu, a). \tag{111}$$

Notice that in the previous equality the supremum can be taken on the whole real line since we are assuming that $\bar{x}(\mu, a) \in RS(\mu, a)$.

Conversely, suppose that $\bar{x}(\mu, a) \in (RS(\mu, a))^c$. We are going to prove the replica symmetric solution cannot hold. By Theorem 1 we know that

$$\lim_{N \to \infty} \bar{p}_N(\mu, a) = \sup_{x \in \mathbb{R}} \varphi(x; \mu, a) = \varphi(\tilde{x}(\mu, a); \mu, a), \tag{112}$$

where $\tilde{x}(\mu, a)$ denotes a point where the supremum is attained. By Remark 5 one can say that $\tilde{x}(\mu, a) \in [-\sqrt{a}, \sqrt{a}]$. Let's consider two cases, first suppose that $\tilde{x}(\mu, a) \in RS(\mu, a)$, then using the result in [27] we have that

$$\varphi(\tilde{x}(\mu, a); \mu, a) = \varphi_{RS}(\tilde{x}(\mu, a); \mu, a) < \sup_{x \in \mathbb{R}} \varphi_{RS}(x; \mu, a), \tag{113}$$

where the last inequality follows from the assumption $\bar{x}(\mu, a) \in (RS(\mu, a))^c$. On the other hand if $\tilde{x}(\mu, a) \in (RS(\mu, a))^c$ it is known [47] that the pressure of the SK model is strictly smaller that its replica symmetric version, therefore

$$\varphi(\tilde{x}(\mu, a); \mu, a) < \varphi_{RS}(\tilde{x}(\mu, a); \mu, a) \leq \sup_{x \in \mathbb{R}} \varphi_{RS}(x; \mu, a). \tag{114}$$

In conclusion, we have just proved that

$$AT(\mu, a) > 1 \quad \Rightarrow \quad \lim_{N \to \infty} \bar{p}_N(\mu, a) < \sup_{x \in \mathbb{R}} \varphi_{RS}(x; \mu, a). \tag{115}$$

$\square$

## 5 Phase Diagram

This section collects the consequences of Propositions 4 and 5 and resumes how the phase diagram in Figure 1 is drawn.

When $a \leq 1$ the condition (19) is not trivial and identifies a curve that lies above $\mu = 1/a$. Below this curve, for $\mu > 1$, the unique stable maximizer of $\varphi_{RS}$ is $\bar{x}(\mu, a) = 0$. The resulting $q(0, \mu, a) \equiv q(0, \mu, 0)$ has to be intended as the stable solution to the consistency equation for the overlap of an SK model at temperature $\sqrt{\mu}$ in absence of external magnetic field, which is known to be RSB for $\mu > 1$. Hence by (100)

$$AT(\mu, a) = AT(\mu, 0) = \mu \mathbb{E} \cosh^{-4}(z\sqrt{\mu q(0, \mu, 0)}) > 1. \tag{116}$$

This in turn implies the replica symmetry breaking in our model. The de Almeida-Thouless red line in the diagram represents the condition $AT(\mu, a) = 1$ and must lie above, or at most coincide with, the curve (19) since it must contain the entire RSB phase. The red region could contain a mixed phase in analogy with the SK model as explained in Remark 6.

From (116) it is also clear that in an RS phase we must have $\bar{x}(\mu, a) \neq 0$ for $\mu > 1$ otherwise $AT(\mu, a) > 1$. Similarly, for $a \geq 1$ and $1/a < \mu \leq 1$, $\bar{x}(\mu, a) = 0$ cannot be the solution to (18) either since

$$q(0, \mu, a) = q(0, \mu, 0) = 0 < 1 - \frac{1}{\mu a}. \tag{117}$$

Contrarily, in the green region, that is replica symmetric by Proposition 5, the unique possible maximizer is $\bar{x}(\mu, a) = 0$ because $\mu \leq 1/a$. Moreover, we have the following

**Corollary 11** (of Proposition 5). *The model is always replica symmetric for any $a \geq 1$.*

*Proof.* Recall that for $\mu \leq 1$ one has trivially $AT(\mu, a) \leq 1$. In addition to that, thanks to Proposition 4 for $a \geq 1$ and $\mu \geq 1 \geq 1/a$ we can always assume (18). Hence

$$AT(\mu, a) \leq \mu \mathbb{E} \cosh^{-2} \left( z\sqrt{\mu q(\bar{x}(\mu, a), \mu, a) + \mu^2 \bar{x}(\mu, a)^2 a} \right)$$
$$= \mu \left[ 1 - q(\bar{x}(\mu, a), \mu, a) \right] = \mu - \mu + \frac{1}{a} = \frac{1}{a} \leq 1. \tag{118}$$

The thesis follows from Proposition 5. $\square$

*Remark* 6. Let us consider $P_\xi = (\delta_{\sqrt{a}} + \delta_{-\sqrt{a}})/2$, or equivalently $\xi_i = \sqrt{a} \tau_i$ with $\tau_i = \pm 1$. In this case, one can gauge away the signs of the variables $\xi_i$'s in (1) by means of the $\mathbb{Z}_2$ gauge transformation

$$z_{ij} \mapsto z_{ij} \tau_i \tau_j, \quad \sigma_i \mapsto \sigma_i \tau_i, \tag{119}$$

obtaining the Hamiltonian

$$\tilde{H}_N(\boldsymbol{\sigma}) = -\sum_{i,j=1}^{N} \left( z_{ij} \sqrt{\frac{\mu}{2N}} \sigma_i \sigma_j + \frac{\mu a}{2N} \sigma_i \sigma_j \right) \overset{\text{D}}{=} -\sum_{i,j=1}^{N} J_{ij} \sigma_i \sigma_j, \quad J_{ij} \overset{\text{iid}}{\sim} \mathcal{N}\left( \frac{\mu a}{2N}, \frac{\mu}{2N} \right). \tag{120}$$

The latter describes an SK model with a peculiar parameterization. To see this it suffices to consider the parameterization [1], namely

$$\beta H_N^{SK}(\boldsymbol{\sigma}) = -\sum_{i,j=1}^{N} J_{ij}\sigma_i\sigma_j, \quad J_{ij} \overset{\text{iid}}{\sim} \mathcal{N}\left(\frac{\beta J_0}{2N}, \frac{\beta^2 J^2}{2N}\right) \tag{121}$$

and to identify $1/\beta J = T/J = 1/\sqrt{\mu}$ and $J_0/J = \sqrt{\mu}a$. This means that if we draw the phase diagram of the model (120) with $\sqrt{\mu}a$ and $1/\sqrt{\mu}$ on the $x$ and $y$ axes respectively we re-obtain the well known phase diagram of the SK model. In this diagram for instance the curves for fixed $a$ are a family of hyperbolas, and among them $a = 1$ corresponds to the Nishimori line. It is then a simple exercise to show that the phase diagram of the SK model redrawn in the parameterization (120) is qualitatively similar to the one in Figure 1, meaning that the same phases are disposed in the same positions. In particular the Nishimori line is the vertical line $a = 1$.

We finally notice that the model studied in [26] can be seen as a special inference problem in a non-optimal setting where the receiver uses his own Rademacher guess to retrieve a binary signal of which he does not know the amplitude.

We conclude the analysis with the study of the behavior of the solution $\bar{x}(\mu, a)$ of the variational problem (22) around the critical point $(\mu, a) = (1, 1)$. By Proposition 4 we have that $\lim_{(\mu,a)\to(1,1)} \bar{x}(\mu, a) = 0$. Notice that the replica symmetric solution $\bar{x}(\mu, a)$ represents the limiting behaviour of the Mattis magnetization when $AT(\mu, a) \le 1$ and it is not identically vanishing iff condition (19) is satisfied. By Proposition 4 and Corollary 11 the above conditions are always satisfied if $\mu a \ge 1$ and $a \ge 1$. Then it holds

**Proposition 12.** *Assuming that $\mu a \ge 1$ and $a \ge 1$ then $\bar{x}(\mu, a)$ is the unique (up to reflection) solution of*

$$\mathbb{E}\tanh^2(Y(x, \mu, a)) = 1 - \frac{1}{\mu a}, \quad Y(x, \mu, a) = z\sqrt{\mu - \frac{1}{a}} + \mu x\xi, \tag{122}$$

*where $z \sim \mathcal{N}(0, 1)$, $\xi \sim \mathcal{N}(0, a)$ are independent Gaussian. Moreover for $(\mu, a) \to (1, 1)$ we have*

$$(\bar{x}(\mu, a))^2 = \frac{(\mu - \frac{1}{a})\left[\frac{1}{\mu} - 1 + 2(\mu - \frac{1}{a})(1 + o(1))\right]}{t(\mu, a)(1 + o(\bar{x}(\mu, a)))}, \tag{123}$$

*where $t(\mu, a) = \mu^2 a \mathbb{E}\left(2 - \cosh\left(2z\sqrt{\mu - \frac{1}{a}}\right)\right)\cosh^{-4}\left(z\sqrt{\mu - \frac{1}{a}}\right)$.*

*Proof.* Clearly (122) holds by Proposition 4. Using a Taylor expansion of $\tanh^2(b + y)$ around $y = 0$ up to order 3 one obtains

$$\mathbb{E}\tanh^2(Y(x, \mu, a)) = \mathbb{E}\tanh^2(Y(0, \mu, a)) + t(\mu, a)x^2 + g(x, \mu, a),$$

where $g(x, \mu, a) = \frac{(\mu x)^4}{4!}\mathbb{E}\frac{\partial^4}{\partial y^4}\tanh^2(y)\big|_{y=y(z,\xi,x,\mu,a)}\xi^4$. Since $|\frac{\partial^4}{\partial y^4}\tanh^2(y)| \le costant$ uniformly on $y$, we have that $g(x, \mu, a) = o(x^3)$. Then one can write

$$\mathbb{E}\tanh^2(Y(x, \mu, a)) = \mathbb{E}\tanh^2\left(z\sqrt{\mu - \frac{1}{a}}\right) + t(\mu, a)x^2(1 + o(x)). \tag{124}$$

The term $\mathbb{E}\tanh^2\left(z\sqrt{\mu - \frac{1}{a}}\right)$ can be represented using Taylor expansion of $\tanh^2(y)$ around $y = 0$ up to order 4 obtaining

$$\mathbb{E}\tanh^2\left(z\sqrt{\mu - \frac{1}{a}}\right) = (\mu - \frac{1}{a}) - 2(\mu - \frac{1}{a})^2(1 + o(1)). \tag{125}$$

Combining (124) and (125) one obtains (123).

$\square$

The previous Proposition and in particular the expansion (123) can be used to obtain the critical behavior of $\bar{x}(\mu, a)$ as $(\mu, a) \to (1, 1)$ with the constraint $\mu a \geq 1$ and $a \geq 1$. As an example fixing $a = 1$ one gets

$$\lim_{\mu \to 1^+} \frac{\bar{x}(\mu, 1)}{\mu - 1} = 1. \tag{126}$$

Analogously if $\mu = 1$ and $a \to 1^+$

$$\lim_{a \to 1^+} \frac{\bar{x}(1, a)}{\sqrt{2}(1 - \frac{1}{a})} = 1. \tag{127}$$

More generally one can consider a family of hyperbolas

$$\mu_\alpha(a) = \frac{\alpha}{a} + 1 - \alpha, \quad \alpha \leq 1 \tag{128}$$

and define $x_\alpha(a) = \bar{x}(\mu_\alpha(a), a)$. Then expansion (123) leads to

$$\lim_{a \to 1^+} \left( \frac{x_\alpha(a)}{\frac{1}{a} - 1} \right)^2 = (\alpha - 1)(\alpha - 2). \tag{129}$$

The critical behavior around $(1, 1)$ of the magnetization along the above directions is therefore the same of the optimal setting [24].

## 6  Conclusions and Outlooks

In this paper we have shown how to solve, in the finite temperature approach, a matrix rank-one estimation problem in a mismatched setting with a Rademacher prior and studied the paradigmatic case when the signal distribution is Gaussian and factorized. For such case, a complete characterization of the phase space has been given in terms of the two order parameters: the Parisi overlap and Mattis magnetization. Our central result, a nested variational principle over a distribution and a real number, can be extended beyond the Rademacher prior assumption, leading to an SK model with soft spins [48] with a Mattis interaction.

We emphasize that our variational principle pinpoints the presence of the replica symmetry breaking phase in a mismatched inference problem. This is expected to have implications on the algorithms usually implemented to retrieve signal components, such as Approximate Message Passing (AMP). Indeed we have observed, with preliminary numerical tests, that in the RSB phase of the model with Gaussian signal distribution ten thousand iterations of AMP are not sufficient to reach convergence: the values of the local magnetizations keep oscillating. On the contrary less than a hundred were enough in the RS phase, thus confirming the picture in Figure 1. As predicted by the state evolution analysis [49] in the RS phases the algorithm is in agreement with the magnetization and overlap given by the consistency equation (18). The rigorous characterization of the AMP convergence, that seems to be related to the de Almeida-Thouless line, is left for future work.

It is interesting to notice that the model studied here is equivalent, through a Hubbard-Stratonovič transformation as done in [50, 51], to a Boltzmann Machine with one hidden analogic neuron linked to a visible layer of neurons in mean field disordered interaction, *i.e.* a non-restricted Boltzmann Machine. Our result extends also to a finite number of hidden

analogic neurons and leads to a model that includes SK and Hopfield terms. In this regard we mention that the SK term can indeed be generated starting from the Hopfield model adding a form of *synaptic noise* [52, 53] (see eq. (8) in [53] in particular) that blurs the interactions, built from the patterns, precisely as in (23).

Finally, we point out that the result presented in this work could be re-framed within the Hamilton-Jacobi approach [54, 55] obtaining an initial value problem with a concave Hamiltonian and the Parisi pressure as initial condition. Our variational principle would then emerge from the Hopf-Lax formula for the solution to such problem.

## Acknowledgements

The authors thank Jean Barbier, Wei-Kuo Chen, Marc Mézard, Dmitry Panchenko and Manuel Sáenz for several fruitful interactions and useful suggestions. Adriano Barra, Francesco Guerra, Jorge Kurchan, Nicolas Macris and Farzad Pourkamali are acknowledged for interesting discussions. The authors acknowledge support from EU project 952026-Humane-AI-Net and RFO University of Bologna funds.

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
