# Peer review of "An inference problem in a mismatched setting: a spin-glass model with Mattis interaction"

_SciPost Physics, doi:SciPost Phys. 12, 125 (2022)_

## Round 2 · Referee Report · Anonymous (Referee 1) · 2021-12-4

Strengths

Rigorous analysis

Weaknesses

Not much physical novelty

Report

Referee report on the paper “An inference problem in a mismatched setting: a spin-glass model with Mattis interaction”.

This paper proposes a rigorous analysis of a variation of the SK model, where in addition to the long-range gaussian couplings, Mattis like couplings and an external field correlated to those are introduced. As the authors show, and previously known in the literature, if the external field is zero, the model can be seen as a toy model for statistical inference (the so-called Wigner spiked model). The authors consider a mismatched setting where the probability of the ground truth is not known by the observer. The authors rigorously establish the phase diagram, without much surprise, they can prove that the solution of the model is provided by Parisi ansatz and they identify the phases of the system. At the technical level, since the Parisi variational free-energy (or pressure according to the authors) is a min w.r.t. the magnetization and a max w.r.t. the overlaps, they need to use an interpolating ‘adaptive’ Hamiltonian that uncouples the Mattis interaction. In this way they can take advantage of the usual machinery for a conventional SK model (without ferromagnetic or Mattis couplings) in a magnetic field. Although the technique was previously known (and its origin duly cited by the authors) I understand that here it is used for the first time outside a Nishimori point. The results of the paper are certainly correct, although physically not really surprising, and the exposition is clear. I feel personally incapable to judge if there is enough mathematical novelty to justify the publication of the paper.

Requested changes

If the editors decide for publication, a detailed caption should be added to figure 1 explaining the lines and the regions of the phase diagram.

  • validity: high
  • significance: ok
  • originality: ok
  • clarity: high
  • formatting: good
  • grammar: excellent

Author:  Francesco Camilli  on 2022-01-31  [id 2135]

(in reply to Report 1 on 2021-12-04)
Category:
answer to question

  • “If the editors decide for publication, a detailed caption should be added to figure 1 explaining the lines and the regions of the phase diagram.” A detailed caption of figure 1 was added. We accordingly modified the paragraph that was after Proposition 4 to avoid repetitions.

Author:  Francesco Camilli  on 2021-12-22  [id 2043]

(in reply to Report 1 on 2021-12-04)
Category:
remark

We thank the referee for the comments. As pointed out, both the solution of the model and its phase diagram are somehow expected from the Physics perspective. Nevertheless, in our paper we give the first rigorous solution, together with the phase diagram analysis, of the Wigner spiked model in a mismatched setting, which is raising increasing interest in the High Dimensional Inference community. One of the main novelties in our paper consists indeed in showing that techniques, such as the adaptive interpolation, that were thought to rely heavily on Bayes optimality can be instead used when the latter is missing. In the mismatched setting one in particular loses the Nishimori identities which are a powerful tool and remarkably simplify the treatment. The variational principle we proved for the free energy finally clarifies with rigor that Replica Symmetry Breaking and lack of Bayes optimality are virtually two sides of the same coin for the Gaussian channels under study. Furthermore, we stress that we were able to draw the phase diagram thanks to a recent rigorous result by Chen (arXiv: 2103.04802) on the sharpness for the de Almeida-Thouless line in presence of centered Gaussian external fields. Though in the Physics community this line is widely accepted as the true separation between RS and RSB phases in the Sherrington-Kirkpatrick phase diagram, from the mathematical point of view, except for the aforementioned result, this is still an open and challenging problem. We expect the presence of RSB phases to also have implications from an algorithmic point of view, since the algorithms designed to retrieve the signal usually work only in RS regimes.

In conclusion, we believe that our work matches the scope of the Journal, since it bridges a gap in mismatched High Dimensional Inference using rigorous tools and it can be a starting point to understand the role of RSB in inference, showing potential for follow-up work and providing new synergies between two existing fields.

If the editors decide for publication, we will be happy to add a detailed caption to Figure 1 as requested.

---

## Round 2 · Referee Report · Anonymous (Referee 3) · 2022-1-6

Strengths

This paper provides a rigorous analysis of SK model with Mattis interaction.

Weaknesses

This paper is hard to understand for non-expert.

Report

In this paper, the authors derive a rigorous proof for a variation of the SK model with Mattis interaction. The main result of the paper is a variational principle for the thermodynamic limit of the free energy of this model. It relies on the adaptive interpolation technique and to the best of my knowledge, it seems to be the first time that this technique is applied outside the Nishimori line. A motivation for the model (with zero external field) is presented in section 3 where the mapping to a toy model for statistical inference with mismatched priors is given. The case where priors match, i.e. the Wigner spiked model, has been recently studied extensively and this extension should be of interest to statisticians. Unfortunately, as it is written, only the mapping is provided and no interpretation or consequences of the main results of the paper are given in this statistical setting. In particular, section 2.1 is very hard to understand for a non-expert. Indeed, I could not make any sense about the paragraph after Proposition 4. It seems that to understand this section, you need to be very familiar with [44].

Before publication, I would recommend a major revision of sections 2 and 3 to make them more accessible.

Minor comments: - in eq (2) the overlap q(\sigma,\tau) is never used. Remove it from section 2 and introduce it when necessary. - for each Theorem or Proposition stated in section 2, I would add a link to the proof in the paper (like Proposition ... is proved in Section ...) - page 9, perhaps say that r_\epsilon will be chosen latter. - page 10 give a reference for Torricelli-Barrow Th.

  • validity: -
  • significance: -
  • originality: -
  • clarity: -
  • formatting: -
  • grammar: -

Author:  Francesco Camilli  on 2022-01-31  [id 2136]

(in reply to Report 4 on 2022-01-06)

  • “in eq (2) the overlap q(\sigma,\tau) is never used. Remove it from section 2 and introduce it when necessary”. The overlap, together with the Mattis magnetization, is one of the two fundamental quantities studied in the paper. Hence we decided to keep where it is. Moreover, now it appears also in eq (30).

  • “for each Theorem or Proposition stated in section 2, I would add a link to the proof in the paper (like Proposition ... is proved in Section ...)”. We added links to the proof sections: “The proofs of Theorem 1, Corollary 2 and Proposition 3 can be found in Section 4.2”, right below prop.3; “The proofs of Propositions 4 and 5 can be found in Section 4.3.” right below prop.5.

  • “page 9, perhaps say that r_\epsilon will be chosen latter”. We added a line on r_\epsilon as requested “ the interpolating function r_\epsilone will be suitably chosen (see Remark 3 below for instance)” Now at page 12.

  • “page 10 give a reference for Torricelli-Barrow Th”. We replaced “Torricelli-Barrow Theorem” with “fundamental theorem of calculus”, on page 13 now.

  • “Unfortunately, as it is written, only the mapping is provided and no interpretation or consequences of the main results of the paper are given in this statistical setting. In particular, section 2.1 is very hard to understand for a non-expert. Indeed, I could not make any sense about the paragraph after Proposition 4. It seems that to understand this section, you need to be very familiar with [44]. Before publication, I would recommend a major revision of sections 2 and 3 to make them more accessible.” We followed the referee’s requests and suggestions and now section 3 is almost doubled in size and a paragraph has been added in the conclusion section. In particular in Section 3, we provide a way to compute the asymptotic mean square error even in the replica symmetry breaking phase and we comment on the relation with the optimal case. Now it is clear how magnetization and overlap contribute to the error in the reconstruction. Concerning Section 6, we added a whole new paragraph (“We emphasize that our variational principle pinpoints the presence of the replica symmetry breaking” up to “ is left for future work.”) to discuss the algorithmic implications of our results, especially those contained in Section 2.1.

In order to make Section 2.1 more accessible and to clarify why we focus on the Gaussian case we added a new introductory paragraph (starting from “Theorem 1 contains a variational representation…” up to “quantitative description of the phase diagram.”). We believe that the conceptual role of the result in arXiv:2103.04802 has now been elucidated. The paragraph after Proposition 4 has been improved and moved into the caption of Figure 1. Its role now is to illustrate the phase diagram of the model using the two order parameters that are involved in the analytical results obtained in Section 2.1.

We thank the referee for the observations for they helped us improving the paper in the interested sections.

---

## Round 2 · Referee Report · Anonymous (Referee 4) · 2022-1-7

Strengths

The paper considers a spin glass model with added Mattis interaction. In a special case (so called $\mu = \nu$ see section 3) the model can be interpreted as an estimation problem in a mismatched setting when the statistician does not have full information on priors. The statistician takes a Bayesian point of view but with assumed priors. Specifically this the Wigner spi iske model: a rank one matrix observed through a noisy gaussian additive channel must be estimated.

Such problems can be reformulated as statistical mechanics spin glass models of SK type with added Mattis interaction. The problem has been treated and basically fully solved in teh matched case (known priors) were it is rigorously proved that the replica symmetric prediction is valid. The matched case enjoys Nishimori symmetry which is extensively used and the solution is in terms of a single order parameter.

The mismatched case has attracted recent attention and the authors give a few relevant references. In their contribution they rigorously show that the replica solution is exact. It involves two order parameters. One expects two order parameters (overlap and magnetisation) because the mismatched case breaks the Nishimori symmetry present in the matched case.

Main contribution:

1) The precise setting here is an assumed signal (the spins) discrete binary and a true signal distributed possibly more generally. Noise is additive gaussian. The setting and result are new and in particular are not covered by previous and/or quasi simultaneous studies e.g ref [46] (this reference and extensions can only deal with gaussian assumed signal and true signal possibly more general).

2) The variational problem is interesting.

3) Proposition 2 generalizes concentration of the Mattis-like magnetization which was previously only shown when the Nishimori symmetry is present. Here this holds out of teh Nishimori line and is crucial for the analysis. Therefore an important result.

4) While I find the result "natural" in the framework of inference, it is not entirely clear to me "why" this still holds slightly more generally (that is for $\mu\neq \nu$). In other words its is nice that Proposition 2 "still" holds.

Weaknesses

None that I can really see. Two questions however:

1) The authors never explain why exactly they do not need the remarkable identities induced by Nishimori symmetry ? How does this compare to previous works were these identities had been used heavily.

2) Maybe I would have liked to see the inference interpretation of the interpolating Hamiltonian equ (28). I guess its easy to work out by readers (if they care).

Report

I certainly recommend publication of the paper in this journal.

Requested changes

0) On page 4 when the adaptive interpolation method is cited please also cite the relevant papers:

The adaptive interpolation method for proving replica formulas. Applications to the Curie–Weiss and Wigner spike models Journal of Physics A: Mathematical and Theoretical, Volume 52, Number 29 Disordered Serendipity: A Glassy Path to Discovery Citation Jean Barbier and Nicolas Macris 2019 J. Phys. A: Math. Theor. 52 294002

and

The Layered Structure of Tensor Estimation and its Mutual Information Jean Barbier, Nicolas Macris, Léo Miolane, arXiv:1709.10368v3 (55th Allerton conference 2017)

1) Ref [27] there exist a short NeurIPS publication and a long version on arxiv. Please cite both.

2) What is the Torricelli - Barrow theorem mentioned on page 10 ? Reference ?

3) In sec 4.2 please reference the relevant literature where the methodology has been used before.

For example for lemma 8 the analysis is I think pretty standard. However the version given here follows almost exactly ( S. B. Korada and N. Macris, "Tight Bounds on the Capacity of Binary Input Random CDMA Systems," in IEEE Transactions on Information Theory, vol. 56, no. 11, pp. 5590-5613, Nov. 2010, doi: 10.1109/TIT.2010.2070131.). This could be stated.

And for the proof of Theorem 1: the method specially the upper bound follows quite closely ref J. Phys. A: Math. Theor. 52 294002 (reference indicated above).

  • validity: top
  • significance: high
  • originality: good
  • clarity: top
  • formatting: perfect
  • grammar: excellent

Author:  Francesco Camilli  on 2022-01-31  [id 2137]

(in reply to Report 5 on 2022-01-07)

Comments on the Weaknesses section

  • “1) The authors never explain why exactly they do not need the remarkable identities induced by Nishimori symmetry ? How does this compare to previous works were these identities had been used heavily. 2) Maybe I would have liked to see the inference interpretation of the interpolating Hamiltonian equ (28). I guess its easy to work out by readers (if they care).” Both questions 1 and 2 were addressed in Remark 1 in Section 4.1.

Requested changes:

  • “0) On page 4 when the adaptive interpolation method is cited please also cite the relevant papers:

The adaptive interpolation method for proving replica formulas. Applications to the Curie–Weiss and Wigner spike models Journal of Physics A: Mathematical and Theoretical, Volume 52, Number 29 Disordered Serendipity: A Glassy Path to Discovery Citation Jean Barbier and Nicolas Macris 2019 J. Phys. A: Math. Theor. 52 294002

and

The Layered Structure of Tensor Estimation and its Mutual Information Jean Barbier, Nicolas Macris, Léo Miolane, arXiv:1709.10368v3 (55th Allerton conference 2017)”

Done

  • “1) Ref [27] there exist a short NeurIPS publication and a long version on arxiv. Please cite both.”

We could not find the NeurIPS reference for [27] (actual [33]), maybe the referee meant the actual [29]?

  • “2) What is the Torricelli - Barrow theorem mentioned on page 10 ? Reference ?” We replaced Torricelli-Barrow theorem with the fundamental theorem of calculus..

  • “3) In sec 4.2 please reference the relevant literature where the methodology has been used before. For example for lemma 8 the analysis is I think pretty standard. However the version given here follows almost exactly ( S. B. Korada and N. Macris, "Tight Bounds on the Capacity of Binary Input Random CDMA Systems," in IEEE Transactions on Information Theory, vol. 56, no. 11, pp. 5590-5613, Nov. 2010, doi: 10.1109/TIT.2010.2070131.). This could be stated. And for the proof of Theorem 1: the method specially the upper bound follows quite closely ref J. Phys. A: Math. Theor. 52 294002 (reference indicated above).”

Page 19, at the beginning of the proof of the old Lemma 8 (now Lemma 10) we added a sentence to clarify this point. We cited the suggested paper ([17]) together with two other spin glass classical references where the method has conceptually originated ([2,9]).

Page 21, we added the sentence “The lower bound follows from…(see [36] for a nice introduction to this method)”, where reference [36] is the requested one.

We take the occasion to thank the referee for its suggestions in improving the clarity on the role of the Nishimori line and for pointing out the references.

---

## Round 3 · Referee Report · Anonymous (Referee 1) · 2022-2-2

Strengths

As I stated in my first report, I am unable of judging about the mathematical originality of the paper.

Weaknesses

The physical results are not surprising.

Report

The other referees seems to be satisfied by the mathematical novelties contained in the paper. The authors changed the paper according to the referees' suggestions. I do not oppose myself to the publication of the paper.

---

## Round 3 · Referee Report · Anonymous (Referee 3) · 2022-2-17

Report

The authors addressed my comments and the comments of the other reviewers. I think the paper is now ready for publication.

---

## Round 3 · Author Response

To the managing editor of the paper “An inference problem in a mismatched setting: a spin-glass model with Mattis interaction” by F. Camilli, P. Contucci and E. Mingione.

We are ready to resubmit a thoroughly revised version of the manuscript where all the points raised by the referees have been addressed. The major revision of Sections 2 and 3, requested by the Anonymous Report 4 of 2022-1-6, has been done. Section 3 in particular has nearly doubled in size, and now contains a more detailed derivation of some consequences of our result in the inferential setting.
Moreover, we added Corollary 2, now at page 4, that is needed for the analysis in Section 3. Section 2.1 contains now an introductory paragraph as a response to the queries in Anonymous Report 4 of 2022-1-6 in order to make it more accessible. Finally, the paragraph that was below Proposition 4, now Proposition 5, has been improved and enriched and moved into the caption of Figure 1.
For each referee's observations and requests we publicly respond with an itemized set of answers.
Overall the manuscript has improved in quality and clarity and we take the occasion to thank the Editorial Board and the referees.

Sincerely,
the Authors.

---

## Round 3 · List of Changes

The changes have been listed in the public replies to each referee's report.

---

## Editorial Decision

published